# Ferroelectricity in graphene nanoribbon devices enabled by collective water molecule dynamics

Muhammad Awais Aslam[1,9], Igor Stanković [1,2,9] ✉, Gennadiy Murastov [1], Amy Carl [3,4], Muhammad Zubair Khan[1], Zehao Song [5], Kenji Watanabe [6], Takashi Taniguchi [7], Alois Lugstein [5], Christian Teichert[1], Roman Gorbachev [3], Raul D. Rodriguez [8] ✉ & Aleksandar Matković [1] ✉

Water is omnipresent in nanoscale systems, yet its collective dynamics and impact on emerging electronics remain poorly understood. Here, we investigate the role of water molecule dynamics in the ferroelectric response of graphene nanoribbon devices. Our findings demonstrate that the collective dynamics of water molecules stabilize the ferroelectric effect. We find that a minimum bi-layer thickness is required for the temperature stability of the ferroelectric effect. In contrast, mono-layer ribbons show a 70% shrinkage of the hysteresis window between 120 and 400 K. Using a combination of electrical transport measurements and molecular dynamics simulations, we conclude that water molecules bridging between graphene nanoribbon layers stabilize the formation of water clusters via intermolecular Coulomb interactions, driving a robust ferroelectric behavior and remnant polarization observed at the device level. This work lays the foundations for exploiting water dynamics in next-generation ferroelectric heterostructures, with direct implications for neuromorphic computing and memory devices.

Water plays essential roles in a wide range of phenomena, from materials and electronics to life itself[1,2]. Understanding the interaction of water with low-dimensional materials has proven essential for nanofluidics[3], energy storage[4,5], water splitting[6], purification[7,8], and water-assisted ferroelectricity[9–11]. Water plays a critical role in tuning material properties at the nanoscale. It turns a challenge into opportunities for nanoelectronics by modifying electronic transport, charge distribution, and doping, making it a critical factor for applications in neuromorphic computing and molecular-level devices, especially when considering the impact of the edge-adsorbed species on graphene nanoribbons (Gr NRs)[12–16].

Graphene nanoribbons with a high edge-to-surface ratio provide an excellent platform to study the influence of water dynamics at the edge-water molecule interface. Previous work showed that water molecules adsorbed at graphene edges in single-layer microdevices encapsulated by hexagonal boron nitride (hBN) can induce ferroelectricity[10], a phenomenon also observed in nanoribbon-based devices[11]. This phenomenon originates from the dipolar nature of water that can result in switching between two polarization states under external electric fields. However, the exact mechanism behind water-induced polarization in Gr NR devices, the impact of field-related dynamics of edge-adsorbed water, the influence of graphene

[1]Chair of Physics, Department Physics, Mechanics, and Electrical Engineering, Montanuniversität Leoben, Leoben, Austria. [2]Scientific Computing Laboratory, Center for the Study of Complex Systems, Institute of Physics Belgrade, University of Belgrade, Belgrade, Serbia. [3]Department of Physics and Astronomy and National Graphene Institute, University of Manchester, Manchester, United Kingdom. [4]Department of Materials and National Graphene Institute, University of Manchester, Manchester, United Kingdom. [5]Institute of Solid State Electronics, TU Wien, Vienna, Austria. [6]Research Center for Electronic and Optical Materials, National Institute for Materials Science, Tsukuba, Japan. [7]Research Center for Materials Nanoarchitectonics, National Institute for Materials Science, Tsukuba, Japan. [8]Tomsk Polytechnic University, Tomsk, Russia. [9]These authors contributed equally: Muhammad Awais Aslam, Igor Stanković. ✉e-mail: igor@ipb.ac.rs; raul@tpu.ru; aleksandar.matkovic@unileoben.ac.at

thickness on water anchoring, temperature-dependent stability, and the interplay of these factors on ferroelectricity remain poorly understood. These knowledge gaps limit the development of next-generation devices based on water-induced ferroelectricity, including applications in radio frequency switches, neuromorphic computing, and memcapacitors[17–21]. This study builds upon the existing knowledge of water-induced ferroelectricity in graphene edges[10,11]. It brings forward new experimental results demonstrating how collective water dynamics are critical to this effect through graphene layer number and temperature dependence. These results are supported by molecular dynamics simulations describing the collective behavior of water at graphene edges. Our twofold experimental and computational approach shows that this ferroelectric effect at the device level in Gr NRs integrated into field-effect transistors (FETs) strongly depends on the graphene layer number, requiring a minimum bi-layer thickness to sustain temperature-stable ferroelectricity. We elucidate the physical mechanism behind these results as driven by the collective behavior of water that induces edge anchoring and interlayer bridging by molecular water necks that enhance intermolecular Coulomb interactions. By investigating the interplay between external fields, temperature, layer number, and adsorbed water at graphene edges, we open the door to future

nanoelectronics and sensor applications leveraging the ubiquitous nature of water confined at the nanoscale[22–29].

## Results

### Observation of the hysteretic response in oxygen-terminated graphene nanoribbon devices

Figure. 1a depicts the schematic of a Gr NR FET on hBN. Para-hexaphenyl (6 P) organic nanoneedles self-assembled on hBN-Gr-hBN heterostructures serve as self-aligned masks[11,30,31]. Reactive ion etching (RIE) selectively removes exposed graphene to produce Gr nanoribbons, as illustrated in Fig. 1b, c; detailed in the Methods section and Supplementary Fig. S1. The resulting nanoribbons have an average width of 20 nm, as estimated from high-resolution atomic force microscopy (AFM) images (Supplementary Fig. S2). Alternative to organic nanostructures, recent studies have shown that directed self-assembly of block copolymers can produce complex arrays of highly ordered sub-10 nm wide nanopatterns[32–34], that can be used to fabricate 2D material nanoribbon FETs[35].

The top and bottom encapsulation with hBN insulates graphene from potential charge-trap sources, ensuring that external interactions primarily occur at the nanoribbon's edges (Supplementary Fig. S3). Organic nanostructure growth is done near thermodynamic

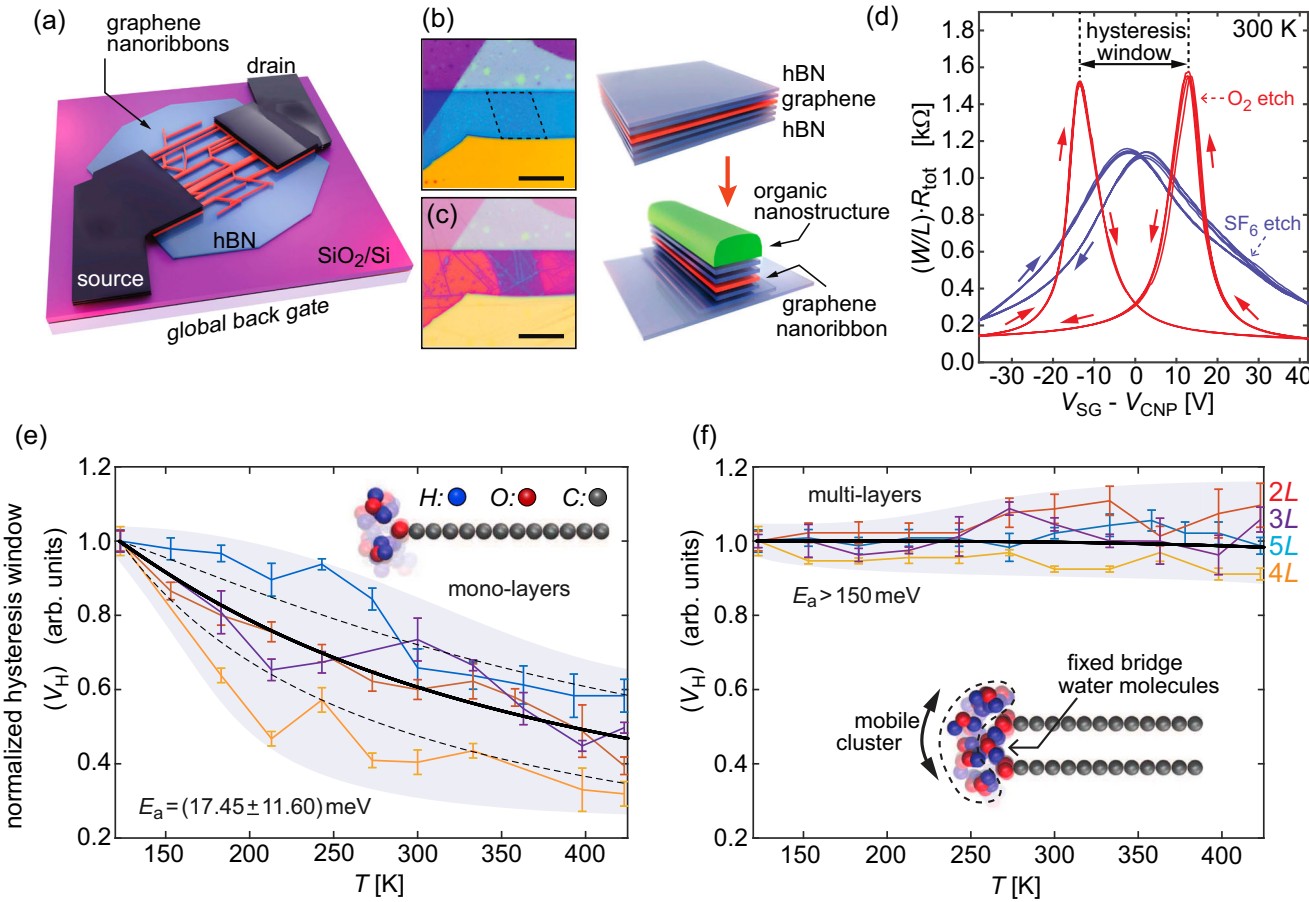

**Fig. 1 | Device structure and the observation of the hysteretic response.**
**a** Schematic representation of a nanoribbon field-effect transistor. **b**, **c** Optical images and corresponding sketches of stacked hBN/Gr/hBN flakes and patterned nanoribbons. **d** Width and length ($W L^l$) scaled total device resistance (two–terminal) transfer characteristics for $2L$ Gr NR FETs (measured under $2 \cdot 10^{-2}$ mbar). Arrows indicate the sweep direction. The $V_{SG}$ range is shifted with respect to the mean $V_{CNP}$ values between both sweeping directions (considering the shifts of 3 V and 28 V for oxygenated and fluorinated ribbons, respectively).
**e**, **f** Temperature dependence of the hysteresis windows ($V_H$) normalized to the

value recorded at 120 K, presenting the data for the mono-layer and multi-layer (2–5 layers, with thickness labelled at the right side for each curve in (**f**)) Gr NR FETs. Different lines represent different devices, and error bars depict $V_H$ variation in the subsequent measurements. Shaded areas serve as a guide to the eye. Solid black lines are the Arrhenius function-based analytical fits used to extract the energy barrier ($E_a$). Dashed black lines in (**e**) correspond to one sigma deviations of the $E_a$ estimate for monolayers. Insets in (**e**) and (**f**) present MD simulation models exhibiting typical local configurations of water clusters in the absence of the external electric field.

equilibrium conditions, meaning that even minor changes in the substrate can significantly impact self-assembly. The interfacial bubbles introduce bi-axial tensile strain in the top hBN layer, and the organic nanostructure self-assembly is hindered in such strained regions. As a result, the nanoribbon network effectively "avoids" these bubbles (Supplementary Fig. S3d–g).

The nanoribbon edge termination is controlled by the choice of precursor gas used for RIE. Using oxygen ($O_2$) or sulphur hexafluoride ($SF_6$) results in oxygen- or fluorine-terminated edges, respectively[10]. Electrical transfer results show that devices with oxygen-terminated edges exhibit pronounced and stable hysteresis, with switchable $p$ and $n$ doping characteristics depending on the gate bias ($V_{SG}$) sweeping direction (Fig. 1d). The electrical response comparison between the nanoribbons and the starting flake FETs is shown in Supplementary Fig. S4. The hysteresis window ($V_H$) is defined as the difference between two charge neutrality points (CNPs), which correspond to the maxima in the total two-terminal device resistance $R_{tot}(V_{SG})$. In contrast, $F$-terminated ribbons show negligible hysteresis due to their hydrophobic nature (Fig. 1d, blue plots), as previously observed for graphene flakes[10]. Given that oxygen-termination enables water molecule adsorption, our focus will be on oxygen-terminated graphene nanoribbons and their water-induced ferroelectricity[10,36,37]. To ensure water molecule adsorption on the oxygenated ribbon edges, Gr NR devices were exposed to ambient conditions (relative humidity of 22−26 % at 298 K) for at least 30 min following the RIE step. This procedure was also applied to the $SF_6$-etched devices to maintain comparability between the two datasets.

We have used a total of 22 devices, comprising 16 $O_2$ RIE-treated and 6 $SF_6$ RIE-treated devices. To illustrate sample-to-sample variation in performance parameters, we present peak apparent linear mobility ($\mu_{lin}$) statistics in Supplementary Fig. S5. Furthermore, $\mu_{lin}(V_{SG})$ curves for both $SF_6$ and $O_2$ etched devices were compared before and after the RIE step, see Supplementary Fig. S6. The assembly of the nanostructure network varies between samples, impacting the scattering of ribbon widths. Consequently, the $V_H$ values show a scatter of about ± 25 % among different devices under comparable operating conditions. This scattering of $V_H$ values is illustrated in Supplementary Fig. S7, as a function of the layer number and the total sweeping range of the gate voltage ($V_{SG-tot}$). Further discussion regarding the relationship between ribbon width and $V_H$ is provided later in the text from the perspective of molecular dynamics (MD) simulations.

The details of the mechanism behind the hysteretic characteristics presented in Fig. 1d are known, as Caridad et al. proposed that single molecules switch between the two states[10]. The external gate fields disrupt the equal probability of water arrangements while inducing a torque on the water adsorbed at the edges. The torque results from Coulomb forces acting on the water dipole, allowing it to cross the energy barrier between the two states *ca.* 25−40 meV[10,38,39].

Interestingly, $R_{tot}(V_{SG})$ hysteresis and its thermal stability exhibit profound differences depending on whether the oxygenated edges of the ribbons are mono-layered or multi-layered. Fig. 1e, f compares the temperature dependence of the normalized hysteresis window for mono-layer and multi-layer (2−5 layers) Gr NR FETs over a temperature range of 120−400 K. The $V_{SG}$ sweeping range was fixed between − 40 V and + 40 V for each $V_H$ value, with a constant sweeping rate of 2 V s$^{-1}$.

In mono-layer Gr NR FETs, the hysteresis window decreases as temperature increases, although significant values persist across the whole temperature range. In contrast, multi-layer devices with two or more graphene layers ($L \geq 2$) show a ferroelectric effect with negligible variations across the investigated temperature range (see Fig. 1f). For multi-layer devices ($2L$ to $5L$), we did not observe any significant differences in the response of the Gr NR FETs with different layer

thickness. However, there is a trend indicating that both $V_H$ and $\mu_{lin}$ tend to decrease as the number of layers increases (Supplementary Figs. S5 and S7).

To estimate the height of the barrier observed in the experiments, we fitted the $V_H(T)$ curves (illustrated by solid black lines in Fig. 1e, f) using an analytical model based on the Arrhenius equation. This model considers the activation energy ($E_a$) or the energy barrier between the two polarization states, expressed as the fitting constant in: $V_H(T) = V_{H0} / (1 - \exp(-E_a/(k_B T))$. Here, $k_B$ is the Boltzmann constant, $T$ is the temperature, and $V_{H0}$ scales the fit to the value obtained at the lowest temperature. For the monolayers, we obtained an activation energy of $E_a = (17.45 \pm 11.60)$ meV. The fits for the individual $V_H(T)$ curves of the mono-layer GrNRN FETs can be found in Supplementary Fig. S8. In contrast, the data for the multilayer devices in Fig. 1f shows an energy barrier higher than 150 meV.

These observations led us to hypothesize that a single water molecule switching between the sides of the graphene plane is responsible for the ferroelectric response in mono-layer devices, leading to a temperature-dependent behavior. When the thermal energy of the molecule exceeds the switching energy barrier, the molecules can thermally transition between states, which explains the temperature-dependent results in Fig. 1e. Additional results regarding the dependence of the hysteresis window on the $V_{SG}$ sweeping range can be found in Supplementary Fig. S9.

## Stability of the hysteretic response: excluding charge trap origins

The direction of the observed hysteresis could originate from the charge traps in the gate dielectric layer. However, the same hysteresis direction could also result from the edge dipoles, as illustrated in Supplementary Fig. S10. To further verify the nature of the observed hysteresis, we performed several experiments to rule out the influence of extrinsic effects such as impurities and charge traps. Since trapping can occur on the timescale of seconds[40,41], we used varying sweep rates (from 0.5 V s$^{-1}$ to 20 V s$^{-1}$) to evidence its effects on $V_H$. Figure 2a shows that $V_H$ as a function of the sweeping rate remains largely unperturbed at 298 K and 77 K under forward and backward gate sweeps, suggesting that the hysteresis is not dominated by charge traps. Furthermore, measurements at 4.2 K in Fig. 2b show that the hysteresis persists, ruling out contributions from thermally activated traps[42,43]. Interestingly, near the charge neutrality point, the resistance exhibits aperiodic fluctuations with high reproducibility, indicating the opening of a transport gap. While such fluctuations could be attributed to quantum interference phenomena[44,45], Chen et al.[46] showed that such insulating states can be attributed to the adsorption of water molecules on the nanoribbon edges, generating strong electric fields that modulate the NRs' band gap.

To further demonstrate the impact of water molecule adsorption, the devices were vacuum annealed at 520 K for 60 min at a pressure of $8 \cdot 10^{-3}$ mbar. Subsequent electrical measurements were performed at 300 K, avoiding any ambient exposure. As shown in Fig. 2c, d, the hysteresis window $V_H$ drastically decreases after annealing, consistent with the desorption of water molecules from the nanoribbon edges. Remarkably, the hysteresis is restored upon exposure to ambient conditions (relative humidity of 22−26 % at 298 K), providing strong evidence for the critical role of water adsorption in the ferroelectric response. In addition, the unintentional $p$−type doping evidenced in Fig. 2c implies that only positive $V_{SG}$ values are required to compensate for built−in charge transfer doping before the total field direction can reverse and switch the water dipoles orientation (Supplementary Fig. S11). The dependence of $V_H$ on the vacuum conditions is detailed in Supplementary Fig. S12, where we observed a 40% increase in $V_H$ when going from vacuum conditions to ambient pressure at 300 K under controlled relative humidity of 40%. Furthermore, $V_H$ remained constant at pressures below $10^1$ mbar.

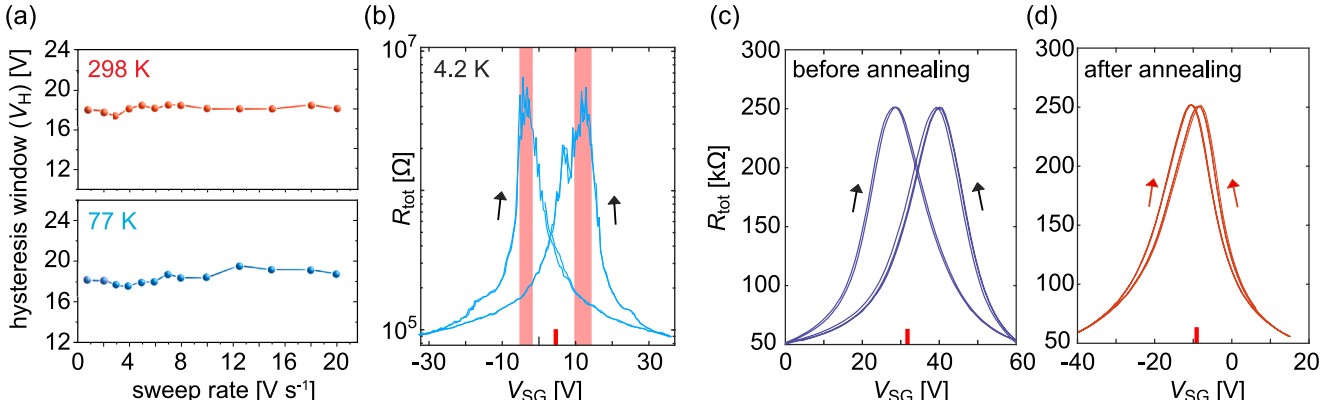

**Fig. 2 | Exclusion of the charge-trap origin of the hysteretic response. a** Change in the $V_H$ with respect to varying $V_{SG}$ sweeping rate (2 $L$ device, measured under $2 \cdot 10^{-2}$ mbar). **b** Total device resistance (two−terminal; semi−log curves) transfer curves demonstrating that the hysteresis prevails at 4.2 K with the emergence of a transport gap, which is highlighted in red color (1 $L$ device, measured under $4 \cdot 10^{-6}$ mbar). **c, d** Total device resistance (two−terminal) transfer curves of the same device before and after annealing at 520 K under vacuum (1 $L$ device, measured under $2 \cdot 10^{-2}$ mbar, 300 K). Red lines next to the $x$−axis in (**b**–**d**) indicate estimates of the unpolarized CNP values. Arrows indicate the sweep direction.

## Molecular dynamics simulations of the oxygenated and water-terminated graphene nanoribbons

We performed MD simulations to understand better the role of water molecules in inducing electric fields and to elucidate the mechanism behind the temperature-dependent hysteresis differences between mono-layer and multi-layer Gr NR devices. The tendency of the water molecules to bind at oxygenated edges and form clusters[39] suggests that a collective behavior stabilizes the molecules, resulting in a temperature-independent effect. Two conditions must be met to achieve such stable ferroelectric ordering of water molecules on Gr edges: (*i*) a low thermal barrier that allows a single polar molecule to switch between states, and (*ii*) a cluster large enough to be stabilized by intermolecular Coulomb interactions. While the first condition is well established, we have designed molecular dynamics simulations to investigate the second condition. The simulations implicitly consider the hydrogen bonds within the dynamically calculated local charges between water molecules and the oxidized graphene edge. Moreover, we also calculate the electric field, which is a direct consequence of the molecule's polarization, as shown later. Our calculations agree with the model presented in ref. 10, and provide new insights into the importance of the collective behavior of water at multi-layered edges.

To conceptualize the mechanistic difference between the two systems, we can think of both mono- and bi-layers as a bi-stable system with two states and a barrier between them. The collective interaction that stabilizes the cluster in one state could essentially be viewed as the deepening of one minimum, depending on how many molecules are in that minimum. This introduces effectively a discrepancy in the rate of returning to the lower occupied state by thermal fluctuations. The process is illustrated in Supplementary Fig. S13.

Therefore, the existence of water interaction and the size of the cluster play a key role in achieving the temperature-independent hysteretic effect. Naturally, there is a limit to how large a cluster can be while remaining anchored at the edge. As the cluster size increases, the induced dipole field also grows. At the same time, the energy required to detach the molecules furthest from the edge decreases (Supplementary Fig. S14). These two effects compete. In the presented simulations, we have limited the amount of water molecules per unit length of the edge to below one-half of the limit determined by the cluster stability in strong external fields at 300 K (more details in the "Methods" section).

The MD simulations of mono-layer graphene confirm the predicted energy barrier of 25−40 meV for switching a single water molecule[10,38]. In contrast, the multi-layer system shows a fundamentally different behavior. The layer separation in graphite allows water molecules to form a bridge between two oxygen-terminated edges. These bridging molecules remain stable and do not react to external electric fields, yet they promote the formation of a water cluster around them. MD simulations show that anchored water molecules within the cluster can sustain high electric fields (up to 5 V nm⁻¹) and temperatures up to 500 K before desorbing from the edges. The cluster surrounding the bridging molecules is mobile in the electric field, and its collective behavior helps stabilize the structure, resulting in a hysteresis window that is essentially temperature-independent. This collective self-stabilizing behavior of polar objects resembles that previously observed in colloidal systems and molecular motors[47,48]. Our calculations do not rule out the formation of water clusters in mono-layers. In mono-layers, flipping of individual molecules across the basal plane of the ribbons is frequently observed, even in the presence of clusters. However, such transitions of individual molecules are essentially absent in the bi-layer case. The distribution of water molecules along the edges in both the 1 $L$ and 2 $L$ cases is presented in Supplementary Fig. S15.

Expanding on our findings, we investigated how varying electric fields affect water clusters using MD simulations. An external homogeneous electric field generated by infinite planes was applied to 1 $L$ and 2 $L$ Gr NRs, as described in the Methods section. The fraction of polarized molecules above ($n^+$) and below ($n^-$) the graphene layer quantifies the ferroelectric effect arising from molecular switching. We observed charge bi-stability and switching between two states under a cyclical change of the external field. The resulting difference between $n^+$ and $n^-$ per unit length of the NR edge is presented in Fig. 3a, b, comparing the hysteresis loops obtained for mono- and bi-layer Gr NRs. It is important to note that in the experiment, the electric field source was generated between the ribbon and the flat surface (gate electrode). In contrast, in the simulations, the electro-neutral system with the ribbon and water molecules was placed between two uniformly charged planar electrodes, generating a homogeneous external out-of-plane field. Further, in the simulations, the electric field applied was over an order of magnitude higher than in the experiment when considering a parallel capacitance model. This was done to reduce simulation time since the electric field strength exponentially prolongs the switching time of the model system. Field enhancement is expected in our experiments due to fringing capacitance effects[10]. An estimate of the enhancement and the field profiles based on a finite elements model of the device structure is provided in the Supporting Information, Supplementary Fig. S16. We anticipate about one order of magnitude enhancement of the out−of−plane field component due to the

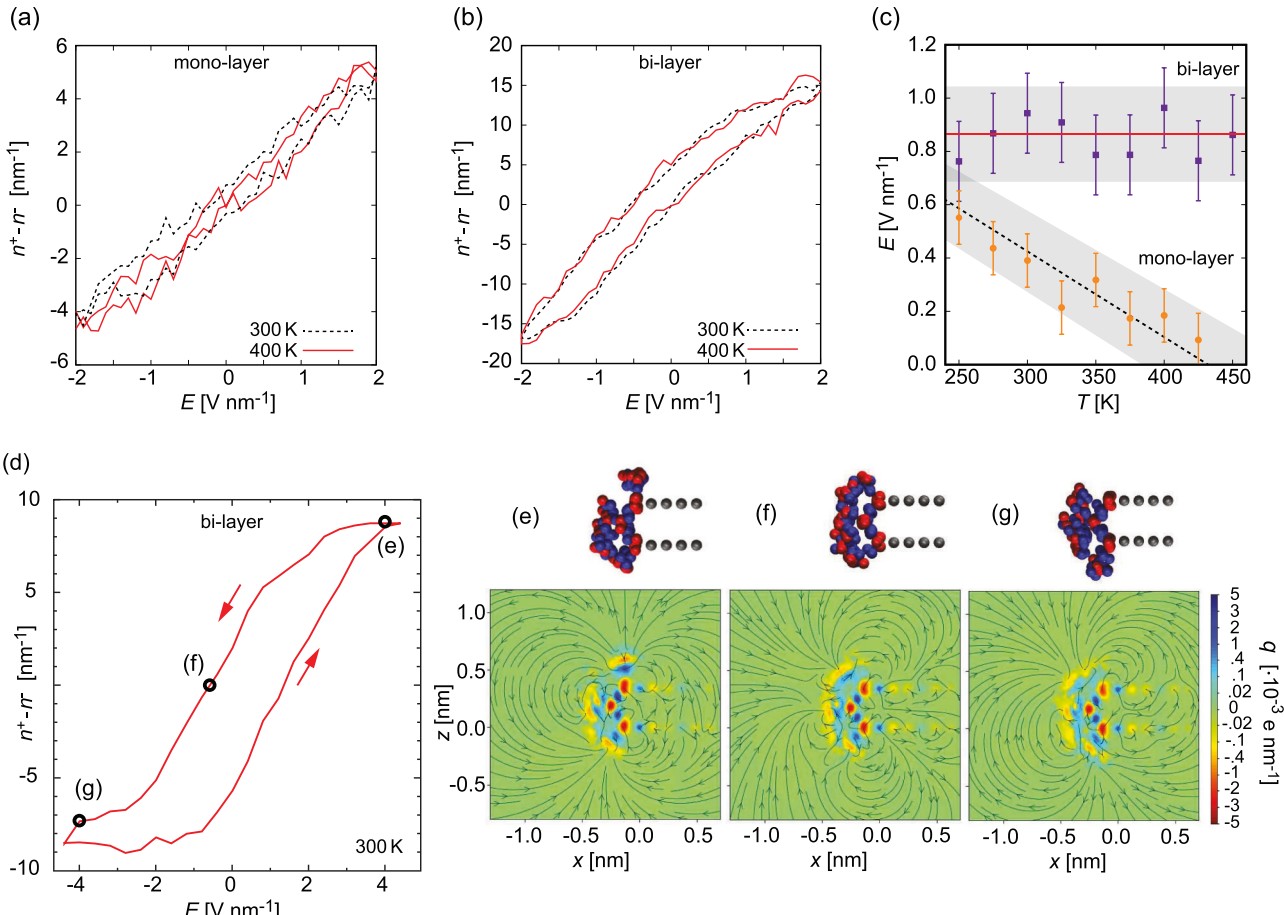

**Fig. 3 | Molecular dynamics calculation results. a, b** Obtained from the MD simulations, respectively, for mono- and bi-layer nanoribbon, the count of the difference of the water dipoles above and below the graphene basal plane ($n^+ - n^-$) per unit of the edge length, as a function of the applied external electric field. Solid red and dashed black hysteresis loops present the responses of the systems relaxed at 400 K and 300 K, respectively. **c** Temperature dependence of the opening of the $n^+ - n^-$ hysteresis as a function of the temperature, considering the external field sweeping range of ±2 V nm⁻¹. Circles indicate mono-layer and squares bi-layer nanoribbons. Solid lines present the mean linear fits, and the dashed area serves as a guide to the eye. Error bars present the standard deviation from five different curves. **d** Example of the bi-layer nanoribbon $n^+ - n^-$ hysteresis with increased external field sweeping range up to ± 4.1 V nm⁻¹. Points labeled (**e**–**g**) in (**d**) have the cross-sections of the resulting model structure presented at the top of the sub-panels (**e**–**g**). The respective density of charges, as well as electric field lines, are shown to scale below. The field in (**e**–**g**) follows a downwards sweep direction from 4.1 V nm⁻¹ to 4.1 V nm⁻¹, and depicts the collective ordering of water at (**e**) $E_{ext} = 4$ V nm⁻¹, (**f**) $E_{ext} = 0.8$ V nm⁻¹, and (**g**) $E_{ext} = -4$ V nm⁻¹. **f** presents the coercivity point, in which the nanoribbon experiences approximately a net zero out-of-plane dipolar field from the water clusters adsorbed at the edges.

fringing capacitance effects, reaching up to 2 V nm⁻¹ within the experimentally achievable biasing range.

For a mono-layer ribbon, a weak hysteresis in $n^+ - n^-$ is observed as a function of the external electric field. Moreover, this hysteresis decreases with increasing temperature (as shown by the orange circles in Fig. 3c), indicating a low energy barrier between the states. This phenomenon can be attributed to the absence of a bridging water molecule found in multi-layer systems. Our experiments with mono-layer Gr NRs showed a more robust hysteresis, especially at elevated temperatures. The discrepancy between the modeled and experimentally observed mono-layer systems may be attributed to more complex interfaces present in the actual experiments that are not captured in the simulations. The model considers a free-standing graphene ribbon, while the experiment involves hBN encapsulation, which likely creates complex electrostatic environments that could also partially enhance the collective behaviour of the adsorbed water clusters. In addition, the model system has a considerably shorter temporal evolution (4 ns) in response to the external electrostatic perturbations compared to the experimental timescale of over 100 ms. This shorter simulation time scale may prevent the model from adequately relaxing within the achievable 4 ns window, especially at

temperatures below 250 K. In contrast, the experimental system has more time to achieve polarization and to experience decoherence of the polarized state once the external field is turned off. Consequently, direct quantitative comparisons of the remnant field values between the experiment and the model are not feasible. Nevertheless, the model effectively captures the trends in temperature and layer number observed in the experiments, allowing us to elucidate the physical mechanisms behind the ferroelectric behavior of Gr NRs devices.

For a bi-layer NR, a prominent hysteresis was noted in the MD simulations (Fig. 3b), and the opening of the hysteresis loop was found to be temperature-independent in the probing range from 250–450 K, as shown by purple squares in Fig. 3c. The field lines and field strength acting on a $2L$ nanoribbon generated via MD simulations are presented in Supplementary Fig. S17. As the electric field decreases, some molecules remain stationary until the field polarization switches, causing these molecules to migrate to the other side with respect to the nanoribbon's basal plane.

The water cluster at the edge of a $2L$ NR is stable in the external electric field up to 4.1 V nm⁻¹, after which the molecules were observed to be removed from the cluster by the external field at 300 K. $n^+ - n^-$ hysteresis presented in Fig. 3d, uses the external field range up to the

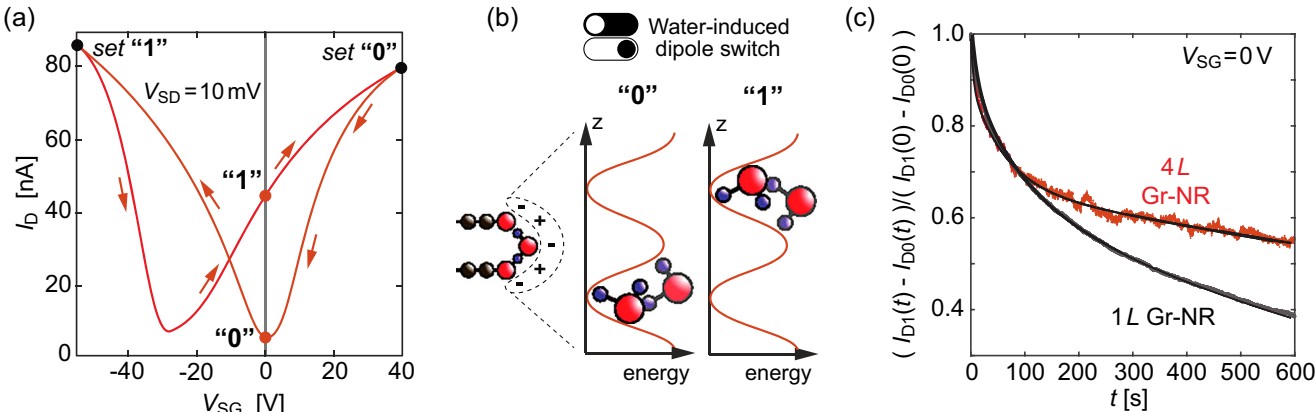

**Fig. 4 | Probing remanence of the hysteretic response. a** The hysteresis curve for $1L$ Gr NR FET after the application of a bias stress (measured under $2\cdot10^{-2}$ mbar; unpolarized position of the CNP $\sim 2$ V). **b** The schematic representation of retention states for the water clusters adsorbed at the edges. **c** The source-drain current deviation in time with respect to the $1L$ $\tau = (1041 \pm 315)$ s, and $4L$ $\tau = (3020 \pm 125)$ s GrNR devices (measured under $2\cdot10^{-2}$ mbar at 300 K).

limit of the desorption of the molecules from the cluster. In this case, a clear indication of saturation is observed. Such high fields probed in these MD simulations are not reachable in the experiment due to the $SiO_2$ and hBN field breakdown. To visualize the ordering of the water cluster, we follow its evolution along a down-sweep of the field in the hysteresis shown in Fig. 3d. Figure 3e–g represents snapshots of the edge segments for a $2L$ Gr NR system with the corresponding charge distribution and induced electric field lines (below). The system consists of both externally applied fields (not shown) and induced fields with opposite polarities in the $z$-direction. Figure 3e presents the case of saturation with an external field of $4$ V nm$^{-1}$ applied in the $z$-direction. An induced charge redistribution is observed due to the molecular polarization at the edges, which in turn creates an effective electric dipole moment. This changes the local field over the whole graphene bi-layer profile. The effective electric field is, therefore, a sum of the two fields, and as the external field decreases, they could cancel each other in parts of the graphene plane. Fig. 3f displays the configuration where the induced electric dipole points roughly along the basal plane of Gr NR, *i.e.*, representing one of the coercivity points of the out-of-plane dipole field hysteresis. Consequently, the induced electric field component perpendicular to the ribbon will be negligible. Finally, for $-4$ V nm$^{-1}$, we observe a complete reversal of the induced electric field (Fig. 3g). As the field strength increases, the migration and molecular polarisation repeat in the opposite order, generating a hysteresis loop. The bonding between the polar molecular ensemble and the graphene edge, together with the intermolecular Coulomb interactions, should be strong enough to prevent the external electric field from tearing off molecules from the cluster (Supplementary Fig. S14b). The details of the large-scale atomistic model are presented in Supporting Information Supplementary Fig. S18.

The width of the ribbons significantly impacts the strength of the dipole-induced fields. Through MD simulations of polarized dipoles, for both $1L$ and $2L$ ribbons, we have extracted the dipole-induced fields perpendicular to the basal plane of graphene and their dependence along the direction of the ribbon width. As shown in Supplementary Fig. S19, the field induced by a single polarized edge follows the theoretically expected $x^2$ dependence. We hypothesize that an optimal width of about 10 nm maximizes the intensity of the hysteresis effect, arising from two competing processes. For ribbons that are too narrow, the field induced by one row of polarized edge dipoles begins to affect the dipoles at the opposite edge, resulting in an antiparallel state. For widths below 10 nm, the field from one polarized edge is expected to impact the other edge, overcoming external fields for widths below 5 nm. In the other extreme, as the ribbon width increases,

the electrical response begins to show more bulk-like properties of graphene, resulting in a reduction of hysteresis. Our MD simulations suggest that for widths above 100 nm, the contribution from the edges to the electrical response becomes negligible.

## Probing the remanence of the water dipole-induced fields in graphene nanoribbons

To confirm the pronounced hysteresis predicted by the MD simulation, we tracked the drain current ($I_D$) evolution of Gr NR FETs without an external gate field after pre-biasing the devices into the $n^+$ or the $n^-$ state. Figure. 4a presents a hysteretic transfer curve of the water-terminated Gr NR FET, where the range of the gate voltages was chosen so that one of the charge neutrality points is at $V_{SG} = 0$ V[49]. The asymmetric $V_{SG}$ sweep approach was used for CNP position adjustment (Supplementary Fig. S20). This way, the difference in $I_D$ for the $n^+$ and the $n^-$ states is maximized. The two $I_D(V_{SG} = 0$ V) states can be defined as "0" and "1", corresponding to the low or the CNP state and the high current state, schematically illustrated for a bi-layer in Fig. 4b.

By sweeping from $V_{SG} = 0$ V to one end of the range and returning to $V_{SG} = 0$ V, the system is set either to the $I_D$ "0" or "1" state. From this point on, the $I_D$ is recorded as a function of time without an externally applied gate field. The dipole-induced remanent field was observed in both mono- and multi-layer Gr NR FETs. The results comparing a mono- and a multi-layer Gr NR FET are shown in Fig. 4c, where the normalized difference between $I_{D1}(t)$ and $I_{D0}(t)$ was tracked for 600 s at 300 K and under low vacuum. To characterize the decay of the field, the curves were fitted to $C\cdot\exp(-t\,\tau^{-1})$ where $C$ stands for a scaling constant and $\tau$ for a time constant.

The remanence of the field for a mono-layer is less pronounced, and the current difference between the two states decays faster than in the multi-layer Gr NR FETs (Fig. 4c). The time constants indicate that the $1L$ Gr NR devices lose 90 % of the initial $I_D$ difference between the "0" and "1" states in about 30 minutes and multi-layer Gr NR FETs in about 90 minutes. Further, a faster decaying component with $\tau = (64 \pm 32)$ s, responsible for about 15–20% of the initial drop of the $I_D$ difference, was noticed in all devices and attributed to weakly bound molecules. Gr NR FETs with $L \geq 2$ require several hours of relaxation to have the initial transfer curve sweep starting from the depolarized state of the water molecules, which is consistent with the extracted time constants.

## Discussion

Our study demonstrates the critical role of water dynamics in stabilizing ferroelectric effects in graphene nanoribbon FETs. A key finding

is the difference in temperature dependency of the hysteresis window between water-terminated mono-layer and multi-layer nanoribbons. In bi-layer or thicker NRs, the ferroelectric response remains essentially temperature-independent up to 450 K due to a collective self-stabilizing effect. This behavior requires a large water cluster, allowing molecules to remain bound in one state through intermolecular Coulomb interactions. Molecular dynamics simulations reveal the mechanism behind the different ferroelectric behavior between mono- and multi-layer NRs. In multi-layer nanoribbons, the presence of two adjacent layers enables the formation of a water molecule bridge that, along with the oxygen-terminated edges, promotes the formation of a water cluster. Our model predicts the collective behavior of this cluster and the bistability in external electric fields, resulting in hysteresis of the induced dipole fields acting on the Gr NRs. The robust temperature-independent hysteresis response, characterized by high remnant fields in water-terminated graphene nanoribbons, makes them promising candidates for the creation of heterostructures with other 2D materials. Improvements using dual gate architectures, high-$k$ dielectrics, and fabricating narrower ribbons could lead to the development of water-switch memristors. In addition, using ribbon networks as electrodes in vertical FET configurations, combined with dipole fields to modulate an atomically thin 2D semiconductor placed over the ribbon network, would enable the creation of reconfigurable memory transistors. These innovations could significantly enhance core functionalities for future computing in memory and synaptic circuits based on water switches.

## Methods

### 2D material heterostructures

Crystals of 2D materials were mechanically exfoliated from bulk material and transferred onto a 290–300 nm $SiO_2$/Si substrate using commercially available Nitto tape. The substrates were chemically cleaned by sonication in acetone and IPA, as well as plasma cleaned in a mixture of oxygen and argon. Mono-layer and multi-layer thick flakes were identified via optical contrast and Raman measurements. Mono-layer graphene crystals were transferred onto hBN substrates using deterministic dry transfer methods. The transfer was achieved using metal-coated $SiN_x$ membranes rather than the conventional polymer stamp transfer techniques[50]. Graphite electrodes and top hBN capping were transferred via a PDMS-based deterministic dry transfer method.

### Organic nanostructures

Organic molecules were grown by hot-wall epitaxy[11]. As a source, parahexaphenyl (6 P) was used. The molecules were sublimated at 493.15 K under $2·10^{-6}$ mbar pressure in a vertically mounted semi-closed quartz tube. The walls of the tube were kept at 513.15 K. The reactor tube is closed by the sample holder that is kept at a temperature of 383.15 K. To ensure minimal temperature variation between the subsequent growths, the system was left to stabilize for 45 min after all three growth temperatures were reached, yielding under ± 1 K variation during the growth. The samples were exposed to the 6 P vapor for 4–6 min, resulting in an equivalent of 0.8 ML (mono-layer) – considering layer-by-layer growth of 6 P on the surrounding $SiO_2$/Si. An example of the nanoribbon network (AFM topography map) after the etching step is presented in Supplementary Fig. S2.

### Reactive ion etching

The etching parameters were adopted from the reference[11]. Reactive ion etching (Oxford Instruments PlasmaLab 100) was done in two steps; first employing $SF_6$ plasma (50 sccm, 80 W, 8-10 s) to remove the top hBN layer, followed by oxygen plasma (50 sccm, 80 W, 4–10 s) to remove the graphene layers and to oxygenate the nanoribbon edges. Alternatively, prolonged $SF_6$ etching (20 s) was used to etch through the whole stack, followed by brief oxygen etching to oxygenate graphene nanoribbon edges.

To ensure that the water molecules can populate the oxygenated ribbon edges, upon etching, all devices were exposed the ambient air, with regulated temperature and relative humidity (298 K and 22–26 %). The waiting time of at least 30 minutes was applied under these conditions for each device in between the RIE step and the electrical measurements under vacuum. Further, no significant difference was observed between the freshly etched devices (with the 30 min holding time) and the devices that were tested, even several days after the RIE step, also stored under ambient conditions.

### Electrical characterization

Electrical characterization of the flake and Gr NR FETs was done using a Keithley 2636 A Source-Meter attached to the Instec vacuum probe station. The samples were contacted with Au-coated Ti electrical cantilever microprobes. Variable temperature (77 - 425 K) electrical measurements were performed using a liquid nitrogen flow cryostat with the devices mounted on a silver plate for thermal uniformity. The temperatures were monitored via Instec's mK2000 temperature controller connected to the probe station with a resolution of 0.01 K. To ensure stable and reproducible electrical measurements, upon reaching the targeted temperature, the system was left to thermalize for 10 min prior to the electrical sweeps. This ensures that during the electrical measurements, the variation of the temperature is below ± 0.5 K. Cryo (4.2 K) measurements were performed using a closed-cycle He cryostat and Lake Shore cryotronics temperature controller. All electrical measurements carried out in the Instec probe station (temperature range of 77–673 K) were performed at $(2.2 ± 0.3)·10^{-2}$ mbar or as specified in the figure captions. The electrical measurements carried out at 4.2 K were done under the pressure of $(4.0 ± 0.1)·10^{-6}$ mbar. To ensure stable low temperature, a holding time of 60 minutes was applied between reaching the targeted temperature and the start of the measurements. This ensured that the fluctuations were below ± 0.1 K at 4.2 K. Sample annealing was done in the Instec probe station under $(8.5 ± 0.5)·10^{-3}$ mbar, for 60 minutes and at 520 K. The annealing parameters were set to remove the leftovers from the organic mask successfully, and not to damage the ribbons or to compromise the gate dielectric. The resistance values are presented as the total device resistance and were measured in a two-terminal configuration. The presented resistance values are neither corrected for the lead resistance nor for the contact resistance between the nanoribbons and graphite electrodes.

Electrical characterization under varied vacuum conditions (Supplementary Fig. S12) was performed by dosing air with a controlled relative humidity level of 40% using a mass flow meter from Omega, with a flow precision of 0.1 sccm (standard cubic centimeters) and a range of 0.1–200 sccm. Vacuum levels were maintained to below 1% variation during the electrical sweeps. To achieve lower pressure levels, the flow of the humid air was set against the continuous pumping. To achieve the pressure levels in the range of $10^2$ mbar, the chamber was disconnected from the pumps, and a small dose of humid air was released to the chamber.

### AFM measurements

AFM measurements were performed using the Horiba/AIST-NT Omegascope AFM system. Nanosensor probes were used (resonant frequency - 300 kHz, tip radius of 2–4 nm). AFM images were processed in the open-source software Gwyddion v2.56. For AFM image processing, zero-order line filtering was applied, leveling of the base plane, and tip deconvolution were considered, using a model tip with a radius of 4 nm.

### MD simulations

The local configuration of $H_2O$ molecules and the binding states at the oxygenated graphene nanoribbon (Gr NR) edges were investigated

using a ReaxFF atomistic model incorporated with molecular dynamics (MD) simulations[51–53]. The model was applied to mon- and bi-layer Gr nanoribbons and includes the elements of the experimental system. The graphene NR is 20 nm wide in the $x$-direction, terminated with oxygen, and 34 nm long in the $y$-direction, with applied periodic boundary conditions. To achieve complete (1:1) edge coverage, we chose 4.7 and 9.4 water molecules per nm length of the edge. To test the influence of one edge cluster on the other, the NR width was varied between 5 nm and 30 nm, considering only a mono-layer Gr nanoribbon. We calculated both in-plane and out-of-plane electric fields generated by the edges for various widths of the graphene band. Based on these calculations, we found that for ribbons wider than 5 nm, the electric fields generated within the cluster become two orders of magnitude larger than the average effective field exerted by one cluster on the other.

The MD simulations were performed using time steps of 1 fs and an NVT thermostat in LAMMPS, a commonly used distributed classical MD code[51]. The simulated water temperature ranged from 250 K to 450 K, depending on the simulation, while the positions of carbon atoms in the ribbon were fixed to maintain a planar structure, similar to the experimental setup. The interatomic forces within the graphene-water system were derived using the appropriate ReaxFF potential by Zhang et al.[52,53]. In addition, a ReaxFF model by Chenoweth et al.[54] was tested, with no qualitative differences observed. Charge equilibration (Qeq) is used in molecular dynamics simulations to calculate the distribution of partial charges on the graphene ribbon edge and water molecules. This distribution changes with time to match changes in the local environment[55].

The homogeneous electric field was varied cyclically between $-2\,V\,nm^{-1}$ and $2\,V\,nm^{-1}$ in the simulations used to evaluate the temperature dependence of the hysteresis width, as these field strengths were estimated to be achievable in the experiment, considering the fringing capacitance effects (see also Supplementary Fig. S16). To achieve the saturation, the field was varied cyclically between $-4.1\,V\,nm^{-1}$ and $4.1\,V\,nm^{-1}$. In all cases, the averages over five cycles are shown.

## Data availability
**Source Data:** The complete dataset generated in this study has been deposited in the Zenodo database[56] Source data are provided in this paper.

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

## Acknowledgements

This work is supported by the Austrian Science Fund (FWF) under grants no. I4323-N36 and Y1298-N, and the TPU Development Program Priority 2030. A.M. acknowledges the support from the ERC Starting grant POL_2D_PHYSICS (101075821). I.S. acknowledges the support of the Ministry of Education, Science and Technological Development of the Republic of Serbia through the Institute of Physics Belgrade and the European Union through the ULTIMATE-I project partner Senzor Infiz doo, grant ID 101007825. Molecular dynamics calculations were run on the PARADOX super-computing facility at the Scientific Computing Laboratory, Center for the Study of Complex Systems of the Institute of Physics, Belgrade. K.W. and T.T. acknowledge support from the JSPS KAKENHI (Grant Numbers 20H00354 and 23H02052) and World Premier International Research Center Initiative (WPI), MEXT, Japan. R.G. acknowledges support from the Royal Society, ERC Consolidator grant QTWIST (101001515) and EPSRC grant numbers EP/V007033/1, EP/S030719/1 and EP/V026496/1. A.C. acknowledges support from EPSRC CDT Graphene NOWNANO, grant EP/L01548X/1. RDR acknowledges the Agrochemical Engineering Innovation and Intelligence Base for Oasis Ecology. Views and opinions expressed are however, those of the authors only and do not necessarily reflect those of the European Union or the European Research Council Executive Agency; neither the European Union nor the granting authority can be held responsible for them.

## Author contributions

M.A.A. and M.Z.K. prepared the devices, carried out experiments, and data analysis under the supervision of A.M. I.S. proposed the underlying mechanism and performed the simulations. G.M. performed the measurements for the field remanence. M.A.A., I.S., R.D.R. and A.M. wrote the manuscript. M.A.A., M.Z.K. and Z.S. carried out etching experiments under the supervision of A.L. K.W. and T.T. provided hexagonal boron nitride crystals. R.G. and A.C. provided monolayer graphene on hexagonal boron nitride. R.D.R., M.Z.K. and C.T. helped in the internal review of the manuscript. A.M. and R.D.R. acquired the main funding for the study. All the authors discussed the results and reviewed the manuscript.

## Competing interests

The authors declare no competing financial and non-financial interests.
