## [Transparent Peer Review file · Nature Communications]

Ferroelectricity in Graphene Nanoribbon Devices Enabled by Collective Water Molecule Dynamics

Corresponding Author: Professor Raul Rodriguez

Version 1:

Reviewer comments:

Reviewer #1

(Remarks to the Author)

The manuscript by Aslam et al. reports on the role played by water on the ferroelectric effect of a graphene nanoribbons device. I think the manuscript is well-written and clearly tackles the water molecule dynamics, although some points should be investigated in more detail. Hence, I recommend revision before publication addressing the following comments.

- I do not understand the Authors' choice to use organic molecules like parahexaphenyl (6P) as masks for their nanoribbons in both this manuscript and their previous work (ref. 11), when more precise self-assembled nanopatterns can be exploited (see for instance Kim et al., Adv. Mater. 2023, 35, 2207338). Is there a specific reason? Do not the Authors believe that a "more ordered" pattern would provide more reliable and comparable devices?

- As per previous entry, a more controlled nanoribbons fabrication would allow to control their width (improving the statistics reported in Figure S2) and the pitch, i.e. the distance between two adjacent nanoribbons. I believe that such a higher control will, in turn, enable to answer questions like: can the width of the nanoribbons assumed to be an additional factor among those considered by the Authors (number of graphene NR layers, temperature...)? Is there a trade-off between the width of the nanoribbon and "bulk-like" effects for which the adsorption of water is negligible?

- The main flaw of the manuscript, in this Reviewer's opinion, is that no description of the ambient conditions is provided. For instance, caption of Figure 1 states that the transfer characteristics for 2L GNR FETs are measured under low vacuum (2×10^{-2} mbar), but it is not reported how the water adsorption has occurred in ambient conditions (humidity?). The only humidity value I found was in the caption of Figure 2. However, due to the importance of this factor in the discussion, I think the Authors should clearly add this information throughout the manuscript.

- Yet again in Figure 1, it not clear the differences (if any) among the devices. In particular the thickness range for multilayer devices is 2-5 layer, cannot the Authors be more specific on the number of devices with 2, 3, ... layers? Moreover in the manuscript devices with $L > 2$ are poorly commented, why?

- The temperature at which the devices were annealed in Figure 2c,d is missing, and again the ambient humidity upon exposure (line 129, page 4) is not reported as well.

- Can the Authors comment and compare the hysteresis window for 1L GNR FET measured at 4.2 K (Figure 2b) and 120-400 K (Figure 1e)? Moreover, it seems that the hysteresis window under low vacuum (11.2 V at 300 K? in Figure S8) is smaller than that in high vacuum (15 V? at 4.2 K in Figure 2b), so do the Authors confirm the observed trend -shrinkage of the hysteresis window- even in vacuum?

- I do not understand why clustering of water molecules is ruled out in single-layer nanoribbons.

- The Authors stated that two conditions must be met to stabilize the observed ferroelectric effect (page 4, lines 142-145). How do the two conditions are competitive to each other? Provided that the clustering of water molecules results into a temperature-independent ferroelectric behavior, can the Authors comment on the marked differences between mono and multi-layered nanoribbons to get the same macroscopic effect? Did the Authors investigate the size of the water molecules cluster on the ferroelectric effect?

Reviewer #2

(Remarks to the Author)

In this manuscript, the authors report that graphene nanoribbon (Gr NR) devices with oxygen-terminated edges exhibit stable hysteresis and suggest that water molecules at the graphene edges induce ferroelectricity. They also note differences in the stability of the hysteresis between mono-layer and multi-layer Gr NR FETs, attributing these differences to collective water molecule dynamics, as simulated using molecular dynamics (MD). I find their work interesting and believe it could have significant implications for future research and applications. However, I am not entirely convinced that the transport measurements specifically reflect the role of water molecules at the edges, while safely excluding the influence of other factors.

Below are some concerns I would like to raise:

1. Could the authors provide the mobility values for their mono-layer and multi-layer Gr NR devices encapsulated by hBN? Given the high mobility typically observed in hBN-encapsulated devices, it would be useful to determine whether the observed hysteresis is primarily related to the Gr NR edges. Additionally, could the authors explain why the mobility of the SF6 RIE device is significantly lower than that of the O2 RIE device in Figure 1(d)?
2. The authors mention an energy barrier of 25-40 meV for switching a water molecule. Can authors estimate the size of this barrier from Figure 1(e)? Is the size of barrier consistent with their observations? The hysteresis shown in Figure 1(e) remains robust up to 420 K, which also appears to differ from their MD model calculations. The authors attribute this discrepancy to complex interfaces in the experiment, but further clarification would be helpful.
3. The authors dismiss the role of charge traps because the hysteresis persists at 4.2 K. However, can the authors rule out the possibility of water-air bubbles at the van der Waals interfaces as a contributing factor?
4. The authors could consider conducting experiments on wider graphene devices encapsulated by hBN. Could they comment on how the width of the graphene affects the hysteresis characteristics?

Reviewer #3

(Remarks to the Author)

This manuscript reports on the role of water molecule dynamics in the ferroelectric response of graphene nanoribbon (Gr NR) devices. Through an analysis of the dependence on the number of graphene layers and temperature, it reveals the stabilizing effect of the collective behavior of water molecules on the ferroelectric effect. To enhance the quality of the manuscript, minor revisions should be considered. All recommendations are listed below.

1. The manuscript involves the preparation of multilayer graphene nanoribbons, water molecule adsorption, and temperature-dependent measurements. However, it does not clarify how experimental consistency was ensured, particularly regarding the reproducibility of experiments under varying temperature and electric field conditions. The authors should demonstrate the consistency of experimental results across different devices.
2. The experimental results in the manuscript show good consistency with the molecular dynamics simulation results for multilayer graphene nanoribbons (Gr NRs), but there are certain discrepancies in the case of monolayer graphene nanoribbons. For example, the opening of the hysteresis window shown in Fig. 3 (a) is much less clear as compared to the experimental results (Figure S4). The authors explained this by introducing the possible more complex interfaces due to hBN encapsulation, please explain in more detail.
3. The time scale of the molecular dynamics simulations (4 ns) in the manuscript significantly differs from the experimental time scale (100 ms). In what way could this substantial discrepancy in time scales affect the accuracy of the simulation results?
4. The ferroelectric effect in multilayer graphene nanoribbons (2-5 layers) remains nearly unaffected at temperatures up to 450 K, whereas the ferroelectric effect in monolayer graphene nanoribbons exhibits significant temperature dependence. While the authors attribute this phenomenon to the absence of collective water molecule behavior in monolayer graphene nanoribbons, they do not explore the mechanistic differences in temperature stability between graphene nanoribbons of varying layer numbers.
5. It would be valuable if the authors could further clarify the potential benefits of designing water-switch-based devices for in-memory computing and synaptic circuit applications.
6. There are minor typographical errors in the manuscript, such as "VH" on line 87 not being italicized. Please ensure that all content adheres to standard formatting conventions.

Version 2:

Reviewer comments:

Reviewer #1

(Remarks to the Author)

The Authors exhaustively replied to all raised concerns and added significant new data and extra discussion in the revised version of the manuscript. I therefore recommend the publication of this manuscript as is.

Reviewer #2

(Remarks to the Author)

Because transport experiments probe the overall average electrical properties of a sample, it is inherently challenging to disentangle contributions that arise solely from the edge. As a result, arguments by authors cannot yet be regarded as completely conclusive. However, it is also true that the authors provided reasonable answers and persuasive arguments to the various concerns I raised.

In my view, the manuscript is suitable for publication at this stage, allowing the broader readership to assess its merits.

Reviewer #3

(Remarks to the Author)

The authors have addressed my comments clearly, I have no more questions.

Point-by-point response to the reviewers' comments and questions

"Ferroelectricity in Graphene Nanoribbon Devices Enabled by Collective Water Molecule Dynamics"
(NCOMMS-23-18909A-Z)

The comments and questions from the reviewers have been separated and presented in regular font, while our responses are provided in blue font, followed by a brief description of the changes made in the revised text, and highlighted in the marked-up version of the revised manuscript and revised SI files, using red font.

Authors' remarks: We sincerely thank all three reviewers for their time and effort in assessing our manuscript, particularly for their positive and encouraging feedback. We have carefully addressed all their comments and questions, which has allowed us to significantly improve the quality and level of our manuscript as detailed in this point-by-point response letter.

--- REVIEWER COMMENTS ---

Reviewer #1 (Remarks to the Author)

The manuscript by Aslam et al. reports on the role played by water on the ferroelectric effect of a graphene nanoribbons device. I think the manuscript is well-written and clearly tackles the water molecule dynamics, although some points should be investigated in more detail. Hence, I recommend revision before publication addressing the following comments.

Authors' response: We thank Reviewer #1 for the positive evaluation of our work. We agree with the reviewer that some points required additional investigations, which, thanks to these constructive comments, helped improve the level and clarity of this work. These revisions include new data and analysis as described below.

R1Q1: - I do not understand the Authors' choice to use organic molecules like parahexaphenyl (6P) as masks for their nanoribbons in both this manuscript and their previous work (ref. 11), when more precise self-assembled nanopatterns can be exploited (see for instance Kim et al., Adv. Mater. 2023, 35, 2207338). Is there a specific reason? Do not the Authors believe that a "more ordered" pattern would provide more reliable and comparable devices?

Authors' response: We thank the referee for discussing the BCP nanopatterning process for 2D materials. While other nanoribbon patterning methods like BCP provide a higher degree of nanostructure order for device optimization, in addition to our experience with 6P masks, we strategically chose these molecules for our fundamental study due to the simplicity of the fabrication process and formation of a clean interface with minimal defects:

1. **Simplicity: The masks leave a bare 2D material surface**, as the molecules do not form a film over the 2D material, but rather strongly de-wet the substrate (resulting in ribbon formation). Therefore, this method requires only one etching step to both pattern the structures and functionalize their edges. This simplicity avoids potential damage to the basal plane and line-edge/width roughness associated with multiple RIE steps.
2. **Clean interface: Parahexaphenyl (6P) interacts weakly with 2D materials.** This method minimizes unintended doping and charge transfer due to mask residues, producing clean interfaces critical to revealing new fundamental phenomena, as demonstrated in this work.
3. **Minimizing interfacial defects:** We use hetero-stacks of 2D materials (hBN/graphene/hBN), which are prone to the formation of interfacial bubbles, even when silicon nitride membranes are used for transfer. Since epitaxial self-assembly of organic molecules is highly sensitive to any perturbations of the substrate's lattice, the **nanostuctures "avoid" assembling on the bubbles** as the commensuration changes due to strain. Consequently, interfacial "bubbles" as a source of potential charge traps are largely avoided in nanoribbon networks. Examples of this are provided within the answer to Question 3 posed by Reviewer 2.

We agree that the precise control offered by BCP is absolutely needed to bring the observed effect to applications.

Changes made: We now discuss in the beginning of the Results section the advances in BCP self-assembly and patterning of 2D materials-based nanoribbons via this method. We also discuss the advantages of BCP for integrating 2D materials-based nanoribbons into future technologies.

R1Q2: As per previous entry, a more controlled nanoribbons fabrication would allow to control their width (improving the statistics reported in Figure S2) and the pitch, i.e. the distance between two adjacent nanoribbons. I believe that such a higher control will, in turn, enable to answer questions like: can the width of the nanoribbons assumed to be an additional factor among those considered by the Authors (number of graphene NR layers, temperature...)? Is there a trade-off between the width of the nanoribbon and “bulk-like” effects for which the adsorption of water is negligible?

Authors' response: This is an excellent point. A narrower width distribution, in addition to a way to control the pitch between the adjacent ribbons, could allow for tuning the hysteresis width and optimize performance with respect to the applied gate bias. However, note that **these parameters should not impact the nature of the observed effect**. Our conclusions regarding the layer number and temperature dependence of the hysteretic effect should remain unchanged, even with a narrower ribbon distribution. We expect that there is an optimal width that maximizes the hysteresis effect. However, performing systematic experiments to elucidate the role of ribbon width on V_H goes beyond the scope of this study.

Following the reviewer's advice, we have estimated the width limit and discussed this issue in more detail in the revised manuscript. From MD simulations of polarized dipoles, both for 1L and 2L ribbons, we have extracted the dipole-induced fields perpendicular to the basal plane of graphene and their dependence along the x direction of the ribbon width. The fields induced by only one polarized edge are illustrated to show that, from the MD model, we obtain the theoretically expected $1/x^2$ dependence (added in Figure S19).

There are two scenarios to consider:

1. As the ribbon width reduces, at some point, the field induced by one row of the polarized edge dipoles would start to affect the dipoles at the other edge. The dipoles from the two edges of a ribbon would effectively try to flip each other, as the edge dipole-induced field has the opposite sign of the external field by the gate that introduces their polarization. From the MD simulations, we estimate that below a 10 nm width, the field induced by one dipole line (edge) starts to affect the polarization of the opposing edge. Below 5 nm, the dipole field would be sufficient to overcome the external field and would likely result in an antiparallel state. Consequently, we expect **a reduction in the hysteresis effect for widths below 10 nm**.
2. Looking into the other extreme, as the ribbon width increases, more “bulk” properties of graphene would be observed in the electrical response, and the hysteresis would reduce. Again, from the MD simulations, we estimate that **the widths above 100 nm GrNRs would give instead a negligible contribution from the edges** to the electrical response. In general, the wider the ribbons, the lower the impact of the edge-induced fields.

Judging from the dipole-induced field distribution from our MD simulations, we expect that the nanoribbon width in the range of **10 nm is optimal for maximizing the hysteresis window** with respect to the total range of the field that can be applied by the gate. Additionally, optimizations with high-k dielectrics, reduced gate thickness, and the use of dual-gate geometries would enable further enhancement of the water switch's effectiveness.

Changes made: The impact of the width of the ribbons is discussed in the main text, and is referenced to the added section in the SI Figure S7 and Figure S19 where, respectively, the statistics of V_H for all the devices (see also the reply to question 6) and additional MD results on the field decay along the ribbon width direction are provided. Estimates of the expected optimal ribbon width, along with other potential improvements of the device architecture, are also discussed in the main text.

Figure S19: MD simulations of the field induced in y -direction (perpendicular to the basal plane) by the polarized dipoles ($E_{\text{ext}} = 2 \text{ V/nm}$ in the positive y -direction, 300 K) considering only one ribbon edge, in the case of 1L ribbon (a-c) and 2L ribbon (d-f). (a,d) Contour maps presenting the surface charge density. Electric field streamlines are superimposed to indicate the direction of electrostatic influence. The mirror-image placement of charges ensures lateral symmetry, and the color-coded regions emphasize strong confinement effects in the near-surface zone. (b,e) corresponding E_y field strength map (semi-log scale). (c,f) Intensity of the dipole-induced fields (solid blue lines) along the x -direction at 0.2 nm above the graphene plane, in log-log scale. A reference x^{-2} trend line (dashed orange) illustrates the asymptotic decay expected from a 1D line of dipoles, confirming consistency with analytical predictions.

R1Q3: The main flaw of the manuscript, in this Reviewer's opinion, is that no description of the ambient conditions is provided. For instance, caption of Figure 1 states that the transfer characteristics for 2L GNR FETs are measured under low vacuum (2×10^{-2} mbar), but it is not reported how the water adsorption has occurred in ambient conditions (humidity?). The only humidity value I found was in the caption of Figure 2. However, due to the importance of this factor in the discussion, I think the Authors should clearly add this information throughout the manuscript.

Authors' response: We agree with Reviewer 1, this is indeed an important omission in the description of the fabrication process that has now been corrected.

In brief, our RIE system (Oxford's PlasmaLab 100) is exposed to ambient air, with a regulated temperature and relative humidity (298 K and 22 - 26 % RH). **A waiting time of at least 30 minutes was applied in these ambient conditions and humidity** for each device between the RIE step and the electrical measurements under vacuum. Further, no significant difference was observed between the freshly etched devices (with the 30 minutes holding time) and the devices that were tested even several days after the RIE step, also stored under the ambient conditions.

Changes made: This is now directly addressed in the main text, and more details are provided in the SI methods.

R1Q4: Yet again in Figure 1, its not clear the differences (if any) among the devices. In particular the thickness range for multilayer devices is 2-5 layer, cannot the Authors be more specific on the number of devices with 2, 3, ... layers? Moreover, in the manuscript devices with $L > 2$ are poorly commented, why?

Authors' response: We are thankful to Reviewer 1 for this question, which allowed us to clarify our idea for multi-layer devices.

We tested thicker devices (up to 5 L) and have not observed any systematic difference between the response of the $L > 2$ FETs. The limit of 5 L was chosen as an empirical estimate to prevent longer RIE steps from damaging the ribbons. During our investigation, we observed that all monolayer devices exhibited a strong temperature dependence of the V_H , whereas all multilayer devices displayed consistent temperature-independent hysteresis. Therefore, **we have grouped the devices to emphasize the fundamental differences between monolayer and multilayer systems**. The goal of Figure 1e,f is to present the evidence of this claim.

After the reviewer's comment, we realized that our strategy needed to be clearly communicated to readers. Additionally, in the revised SI, we specify the exact number of devices used in our study and include statistical graphs comparing the apparent peak mobility and V_H values across all devices (see also the reply to question 6 and reviewer 2 question 1).

Changes made: We have edited the main text and Figures 1 and 4 to clearly indicate the number of layers. We also ensure that within the SI the layer number is clearly communicated for all presented results.

R1Q5: The temperature at which the devices were annealed in Figure 2c,d is missing, and again the ambient humidity upon exposure (line 129, page 4) is not reported as well.

Authors' response: We thank the reviewer for noticing this omission. The annealing was performed at **250°C for 60 minutes** under 8×10^{-3} mbar vacuum. Re-exposure to ambient relative humidity (22-26 %) is now also specified.

Changes made: The humidity and the annealing parameters are now clearly stated in the main text, in the SI Methods, the caption of Fig. 2, and in the SI Fig. S11.

R1Q6: Can the Authors comment and compare the hysteresis window for 1L GNR FET measured at 4.2 K (Figure 2b) and 120-400 K (Figure 1e)? Moreover, it seems that the hysteresis window under low vacuum (11.2 V at 300 K? in Figure S8) is smaller than that in high vacuum (15 V? at 4.2 K in Figure 2b), so do the Authors confirm the observed trend -shrinkage of the hysteresis window- even in vacuum?

Authors' response: We thank the reviewer for this insightful point. Our results show that qualitatively, at 4.2 K, the device shows a robust hysteresis consistent with the trend in Figure 1e, taken as the low-temperature limit where the thermal energy is insignificant to cause depolarization. A direct quantitative comparison between these two experiments, shown in Figure 2b and the SI Figure S12 (originally Fig. S8), is not feasible since the data were obtained from two different devices in two separate measurement systems. The need for different devices arises from the varying environmental conditions required for each experiment. Further, V_H variations are experienced between devices, primarily due to differences in ribbon width. Therefore, we have refrained from making direct comparisons of the absolute values between the different devices.

To demonstrate this and **provide a clear overview of the V_H scattering**, we have added a new statistical summary of the hysteresis window for the entire set of 22 devices used in this study (16 O_2 etched and 6 SF_6 etched). These additions are now discussed in the revised manuscript.

The data presents the hysteresis window over the total gate sweeping range (V_H/V_{SG-tot}) as a function of the number of layers (Fig. S7a), V_H/V_{SG-tot} as a function of V_{SG-tot} (Fig. S7b), and also unscaled V_H as a function of V_{SG-tot} (Fig. S7c).

To answer the question about the vacuum trend, we have carried out a new experiment where we investigate the dependence of the V_H on a single device from the vacuum level (8×10^{-4} mbar) to ambient conditions (40% RH). These new results, presented in Figure S12, show that the V_H reduces by about 40% when going from the ambient pressure down to about 10^1 mbar, and we did not observe any further reduction of the V_H as the pressure was decreased further down to a high vacuum. This definitely **confirms that the hysteresis window shrinks when the ribbons are exposed to the vacuum conditions and that it remains stable below 10^1 mbar**. The apparent trend noted by the reviewer is likely due to device-to-device variability, not a continuous pressure-dependent effect. This question is now discussed in the revised manuscript text.

Changes made: Along with the discussion on the potential ribbon width effects on the V_H , in the revised manuscript we also refer to the V_H variations within our device set (added SI Figure S7), we discuss that the scattering likely originates from the variations in the ribbon widths, and also argue why only relative V_H values are compared between different devices.

Figure S12 (original version SI Figure S8) has been replaced with more detailed experiments that show how the vacuum level impacts the V_H in a range from 8×10^{-4} mbar up to the ambient.

Figure S7. Statistics of the hysteresis window (V_H) considering the entire set of devices used in the study (16 O_2 RIE and 6 SF_6 RIE FETs). Red circles represent O_2 etched devices and purple diamonds represent SF_6 etched ones. Shaded areas serve as a guide to the eye to indicate the trends and the device-to-device scattering of the parameters. The data are extracted from the electrical transfer curves measured at 300 K and under $2 \cdot 10^{-2}$ mbar. Each data point with its error bar is extracted as a mean and standard deviation from five transfer curves. (a,b) V_H is scaled by the total V_{SG} sweeping range (V_{SG-tot}), as a function of the number of graphene layers (a) and as a function of V_{SG-tot} (b). (c) presents $V_H(V_{SG-tot})$ without scaling. Likely, the main source for the device-to-device scatter is the variation in the width of the nanoribbons in the self-assembled networks.

Figure S12: (a) electrical transfer curves (total device resistance; two-terminal) of a 1L Gr NR FET measured at 300 K in ambient pressure and 40 % relative humidity (RH), and immediately after under a vacuum of $8.3 \cdot 10^{-4}$ mbar. (b) for the same device, the dependence of the V_H on the vacuum conditions of the electrical measurement chamber. For each data point, V_H was extracted from the electrical transfer curves measured at 300 K with V_{SG} sweeping between -40 V and +40 V at a 2.5 Vs^{-1} rate. Vacuum level was controlled by adjusting the inlet of 40% RH air using a 0.1-200 sccm flow controller. The measurements were conducted in two ways: starting from the ambient and lowering the pressure, and starting from the lowest pressure and increasing to the ambient. Error bars indicate a standard deviation of the V_H from three subsequent sweeps. Going from 10^{-3} mbar to the ambient pressure and RH 40% the V_H opening increases by about 40 %, and the noticeable increase in V_H starts only after 10^1 mbar pressure.

R1Q7: I do not understand why clustering of water molecules is ruled out in single-layer nanoribbons.

Authors' response: We thank the reviewer for pointing out that this issue was not clearly discussed in the original manuscript. We did not intend to rule out clustering in mono-layers, but rather to highlight the fundamental differences in the nature of the dynamics that give rise to clustering. To clarify, although less frequently than in bi-layers, **water clusters do form in mono-layers**; however, the complex water dynamics involve motion on the graphene basal plane and flips of individual molecules, which make water clusters less stable. In contrast, in bi-layers, the motion of individual molecules across the basal plane is less frequent, while individual molecule flipping was not observed, resulting in larger and more stable clusters. The reasons for these differences are further discussed in our response to the following comment below (R1Q8).

Changes made: This point is clarified in the revised manuscript, now stating that the formation of clusters and the collective motion of water molecules are not excluded for the mono-layers. Additional data on the cluster size statistics per unit length of the edge are provided in the SI Figure S15.

Figure S15. Probability of water cluster density for (a) mono-layer ribbon, and (b) bi-layer ribbon. The data is extracted from a relaxed configuration at 300 K and without applying the external field. The results demonstrate two points: first, the formation of water clusters cannot be ruled out for the mono-layer ribbons, and second, individual water molecules adsorbed at the ribbon edges were essentially not found in the case of the bi-layers. Please, mind that the bars below 4 nm^{-1} correspond to the individually adsorbed molecules, and that for the bi-layers only the mobile water molecules are counted, not the “bridge” molecules between the layers.

R1Q8: The Authors stated that two conditions must be met to stabilize the observed ferroelectric effect (page 4, lines 142-145). How do the two conditions are competitive to each other? Provided that the clustering of water molecules results into a temperature-independent ferroelectric behavior, can the Authors comment on the marked differences between mono and multi-layered nanoribbons to get the same macroscopic effect? Did the Authors investigate the size of the water molecules cluster on the ferroelectric effect?

Authors' response: These two conditions are not directly competitive but regard two characteristics of the ferroelectric effect. In a simplified picture, we can think of both mono- and bi-layer cases as being a bi-stable system with two states and a barrier between them. The barrier to cross between the two sides of the ribbon (the two energy states) is rather similar for both cases, that is the **1st condition – low thermal barrier**. This condition allows the water dipoles to be switched by the gate at the device level. The **2nd condition – collective self-stabilizing cluster** – could be seen as a deepening of one minimum, depending on how many molecules are in that minimum state. This introduces a discrepancy in the rate of returning to the lower occupied minima by thermal fluctuations (see added Fig. S13), and explains how the dipoles remain in a polarized state after switching, resulting in a memory effect.

Therefore, the existence of water interaction and the size of the cluster play a key role in achieving the temperature-independent hysteretic effect. There is, of course, a limit on how large the cluster can be and remain anchored at the edge. The larger the cluster, the larger is the dipole induced field (results added to Fig. S14a), and at the same time, the lower the energy needed to remove the individual molecules at the cluster's periphery since they are less strongly bound (results added to Fig. S14b). These two effects are competitive with each other.

Regarding the differences between mono- and multi-layers, the critical point is how each system supports water clusters. In multi-layers, the water clusters are stabilized by strong interlayer bridging water molecules acting as a stabilizing scaffold. This interlayer anchoring effect enables the formation of larger and more robust water clusters, resulting in the collective temperature-independent behavior observed in our experiments. The interlayer anchoring effect is absent in monolayers (there are no adjacent layers to build the stabilizing scaffold), which limits the size and stability of the water clusters that can be formed. Therefore, in contrast to multi-layers, monolayers are dominated by the dynamics of individual water molecules, resulting in temperature-dependent effects.

Our MD calculations presented in Figure 3 of the manuscript used the largest fully stable clusters with respect to the entire temperature and the external field ranges (up to 4.1 V/nm). Under these conditions, the largest stable clusters at the ribbon edge correspond to 15 water molecules/nm for mono-layers and 20 molecules/nm for bi-layers. Under lower fields and lower temperatures, larger clusters are expected.

Changes made: This issue is now clearly addressed in the revised manuscript. Two figures have been added to the SI. Fig. S13 illustrates the impact of the two conditions on the switching process and the barrier height. Fig. S14 provides additional calculation details on the relation between the amount of water per unit length of the ribbon edge and the stability of water molecules within the cluster at high external fields.

Figure S13. Mechanistic illustration of the self-stabilizing effect on the bi-stable system. (a) Potential energy profile (E_p) of a bi-stable system in equilibrium (equal distribution of the particles between the two states), with the exchange rate between the two states described by $k(T)$, and the energy barrier between the two states (E_a). (b) If the external field (E_{ext}) is applied, the potential profile is perturbed and one minimum is favored. (c) Upon canceling the E_{ext} , the capability of the system to stay polarized depends on the ratio between the barrier height and the thermal energy of the particles. In (a-c), we do not consider interaction between the particles, and even if one state is more occupied, the rate of crossing between the states is the same. In contrast, (d-f) considers that the interaction between the particles favors the state with higher occupation, *i.e.*, reduces the exchange rate. This would be the case when collective dynamics self-stabilizes the system in a polarized state. Let's assume that by thermal fluctuations one particle crosses over the barrier (d) and introduces non-equal occupation of the states. The interaction between the particles and the discrepancy in the occupation between the two states introduces the energy difference between the two minima ΔE and more populated state is slightly more favored (e). Lastly, if we would polarize such an interacting system in an external field (as illustrated in (b)) and switch the external field off, being significantly more populated, one state would be significantly more favored (f). In this case, the exchange rates between the two states would significantly differ, and to de-polarize the system, a much larger effective barrier would be observed.

Figure S14. (a) Calculated strength of the dipole-induced field at the middle of 20 nm wide ribbons, as a function of the amount of water molecules present in the simulation and expressed per unit length of the edge. The model is relaxed in the external field of 2 V/nm in order to polarize the dipoles. (b) Number of water molecules that have detached from the edge after 0.5 ns of the simulation in a high external field of 4 V/nm, shown as a function of the number of water molecules per unit length of the edge. Circles and crosses represent the calculation results for 1L and 2L ribbons, respectively. For both sets of calculations, the temperature was set to 300 K.

Reviewer #2 (Remarks to the Author):

Reviewer #2 comments: In this manuscript, the authors report that graphene nanoribbon (Gr NR) devices with oxygen-terminated edges exhibit stable hysteresis and suggest that water molecules at the graphene edges induce ferroelectricity. They also note differences in the stability of the hysteresis between mono-layer and multi-layer Gr NR FETs, attributing these differences to collective water molecule dynamics, as simulated using molecular dynamics (MD). I find their work interesting and believe it could have significant implications for future research and applications. However, I am not entirely convinced that the transport measurements specifically reflect the role of water molecules at the edges, while safely excluding the influence of other factors.

Authors' response: We are thankful to Reviewer 2's constructive comments; we agree that excluding the influence of other factors is critical to this work. To address this crucial point, we highlight four key experimental observations that collectively demonstrate the role of edge-adsorbed water in the ferroelectric behavior of Gr NR FETs.

1. The hysteresis effect appears only in devices with hydrophilic, oxygen-terminated edges. Identical devices with hydrophobic edges obtained with SF₆ RIE, with all the other parameters kept the same (fabrication steps, graphene thickness, hBN capping, stacking methods, graphite contacts, exposure to ambient humidity), show negligible hysteresis. This points to an effect related to the edge's ability to adsorb water.
2. Vacuum annealing at 520 K completely removes the hysteresis behavior in oxygen-terminated devices. However, when the same device is re-exposed to ambient humidity and measured again under vacuum, we see the reappearance of the hysteretic behavior, proving the direct role of water adsorption from the air.
3. Potential contributions from trapped charges due to the formation of interfacial bubbles are avoided in our fabrication method with self-assembled masks. This is confirmed by new AFM results in Figure S3.
4. Other contributions due to environmental exposure of the nanoribbon basal planes are minimized by hBN encapsulation of the top and bottom sides of the graphene channel. This device configuration isolates the device from environmental charge traps, ensuring that the edges and not the basal planes are selectively exposed for interaction with the environment.

These independent observations provide strong evidence to support our conclusion that the ferroelectric effect observed originates from the dynamics of water molecules at the Gr NR edges instead of other potential artifacts.

R2: Below are some concerns I would like to raise:

R2Q1: Could the authors provide the mobility values for their mono-layer and multi-layer Gr NR devices encapsulated by hBN? Given the high mobility typically observed in hBN-encapsulated devices, it would be useful to determine whether the observed hysteresis is primarily related to the Gr NR edges. Additionally, could the authors explain why the mobility of the SF₆ RIE device is significantly lower than that of the O₂ RIE device in Figure 1(d)?

Authors' response: Following the reviewer's advice, we performed a comprehensive mobility analysis for all 22 devices used in this study. The data are now included in the revised SI. The following results have been added:

- Statistics of the apparent electron mobility (not excluding potential contributions of the contact resistance) at 300 K and for the total of 22 devices used in the study (6 SF₆ etched and 16 O₂ etched) are now added to the SI Figure S5. The graph provides the peak mobility values from $\mu(V_G)$ calculated using the parallel capacitance model. The model is chosen to provide a straightforward comparison between starting flake FETs and Gr NR FETs. If the fringing capacitance model is used (1D wire over a plane), about 1 order of magnitude lower mobility values are obtained for the Gr NR FETs. More details on the fringing capacitance model are available in [[10.1103/PhysRevLett.106.256801](https://doi.org/10.1103/PhysRevLett.106.256801)] [[10.1038/s41699-022-00356-y](https://doi.org/10.1038/s41699-022-00356-y)].
- Several representative $\mu(V_G)$ graphs have been added in the SI Figure S6, two O-terminated and two F-terminated devices, presenting the data for 1L and 2L Gr NRs in each case. $\mu(V_G)$ curves are provided before and after the RIE step for the same device.

Regarding the SF₆ RIE devices and mobility, we conducted a negative experiment with six SF₆ RIE devices. Since the comparison between O₂ and SF₆ RIE was previously reported for the edges of graphene flakes [[10.1021/acs.nanolett.8b00797](https://doi.org/10.1021/acs.nanolett.8b00797)], a larger data-set on SF₆ RIE devices was unnecessary.

Our statistical analysis in Figures S5 and S6 shows that there is no **systematic mobility reduction in SF₆ RIE devices compared to O₂-etched ones**. Since these devices have been used only as a negative experiment, we preselected the lower-performing devices for the SF₆ RIE process. These results confirm that the main difference between the two device types is the presence or absence of water-induced hysteresis, O₂ vs. SF₆-etched, and not their carrier mobility.

Regarding the O₂ RIE devices and mobility: Considering the comparisons between flake- and NR-FETs, and using a parallel capacitance model, $\mu(V_G)$ values were found to be in a similar range as for the original flakes. An increase in peak $\mu(V_G)$ values is commonly observed for O₂ RIE-FETs. Further, $\mu(V_G)$ curves were found frequently to be asymmetric for electrons and holes around CNP+/CNP-, and also exhibit a sweep direction dependency. We attribute this to the interplay between the gate and the induced dipole fields, which can enhance the apparent mobility on one side of the CNP depending on the sweep direction, correlating the electrical characteristics to the dipole switching.

Changes made: Mobility values for the entire device set used in this study and gate-dependent mobility curves are added to the SI Figure S5 and Figure S6. These new results and the difference between the parallel plate and fringing capacitance models are discussed in the revised text.

Figure S5: Device statistics for the linear apparent electron mobility as a function of the number of graphene layers $\mu_{lin}(L)$. The values are extracted as peak mobilities from the electrical transfer curves recorded at 300 K ($2 \cdot 10^{-2}$ mbar). The total of 22 devices that were used in the study are presented. Hexagons correspond to the flake-FETs before the RIE step (step 5 in the fabrication scheme – Figure S1), diamonds represent Gr NR-FETs etched using SF_6 RIE, and circles using O_2 RIE. Mobilities are extracted in all cases using a linear regime approximation and a parallel capacitance model. Fringing capacitance model for the NR-FETs results in about one order of magnitude lower values [S2]. Length and width of each device were determined via a combination of the optical micrographs and AFM topography images. The larger uncertainty for the Gr NR-FETs originates from the uncertainty of the NR widths measured by the AFM.

Figure S6: Linear apparent electron mobility as a function of gate voltage. Each sub-panel compares the $\mu_{lin}(V_{SG})$ curves of devices before and after the RIE step (labeled as flake- and NR-FETs, respectively). The x-axis is compensated for the V_{CNP} values for better comparison. The curves are obtained from the electrical transfer data recorded at 300 K ($2 \cdot 10^{-2}$ mbar) using the linear regime parallel capacitance mobility model. (a) and (b) presents two devices with O_2 RIE etching of 1 and 2 graphene layers, respectively. (c) and (d) present comparable devices to (a) and (b), only with SF_6 RIE step.

R2Q2: The authors mention an energy barrier of 25-40 meV for switching a water molecule. Can authors estimate the size of this barrier from Figure 1(e)? Is the size of barrier consistent with their observations? The hysteresis shown in Figure 1(e) remains robust up to 420 K, which also appears to differ from their MD model calculations. The authors attribute this discrepancy to complex interfaces in the experiment, but further clarification would be helpful.

Authors' response: We thank the reviewer for this multi-part question that led us to perform a new analysis that significantly strengthened our conclusions. Following the reviewer's advice, we estimated the activation energy (E_a) by fitting our experimental data to an Arrhenius model: $V_H/V_{H0} = 1 - \exp(-E_a/k_B T)$. Here, k_B is the Boltzmann constant, T is the temperature, and V_{H0} simply scales the fit to the value obtained at the lowest temperature (considering the experimental range). From the $V_H(T)$ experiments presented in Figure 1e of the original manuscript, for the mono-layers, we have obtained $E_a = (17.45 \pm 11.60)$ meV. The data for the multi-layers (Figure 1h) yields a barrier higher than 150 meV.

Concerning the discrepancy with the MD model and clarification of "complex interfaces", the reviewer correctly noticed that the experimentally deduced hysteresis values seem to be more robust than the obtained ones in our idealized MD model, showing a weaker hysteresis. We attribute this discrepancy to several factors present in the experiments (our "complex interfaces" case) that are not introduced in the MD simulations. First, the hBN encapsulation in our experiments can provide an additional electrostatic potential that contributes to the water cluster stabilization. In contrast, this effect is not considered in the simulations that are based on an isolated Gr NR. Second, in comparison to the nanosecond time scale in which the simulations were run, the experimental values were averaged over a much longer time scale (>100 ms), allowing the water clusters more time to relax and to settle into a stable polarized state, resulting in a more pronounced hysteresis. Following the reviewer's advice, this point is now discussed in the revised text.

Changes made: The estimation of the energy barrier is briefly discussed in the text, the fits and the E_a values are given in Figure 1e,f. Further, Figure S8 is added to the SI, showing separately the fits for each of the mono-layer devices.

Figure 1: (e,f) Temperature dependence of the hysteresis windows (V_H) normalized to the value recorded at 120 K, presenting the data for the mono-layer and multi-layer (2 to 5 layers, with thickness labelled at the right side for each curve in (f)) Gr NR FETs. Different lines represent different devices, and error bars depict V_H variation in the subsequent measurements. Shaded areas serve as a guide to the eye. Solid black lines are the Arrhenius function-based analytical fits used to extract the energy barrier (E_a). Dashed black lines in (e) correspond to one sigma deviations of the E_a estimate for mono-layers.

Figure S8. Separately fitted, Arrhenius equation-based model of the energy barrier height for each of the mono-layer devices from Figure 1e of the main manuscript. The fitting is carried out by the root mean square minimization, and the uncertainties imply a standard deviation.

R2Q3: The authors dismiss the role of charge traps because the hysteresis persists at 4.2 K. However, can the authors rule out the possibility of water-air bubbles at the van der Waals interfaces as a contributing factor?

Authors' response: We thank the reviewer for noticing this critical point that was not carefully addressed in the original manuscript. Indeed, we can rule out contributions from interfacial air bubbles, which is one of the benefits of our fabrication process.

The organic nanostructure self-assembly masks obtained by hot-wall epitaxy (HWE) growth near thermodynamic equilibrium are highly sensitive to substrate strain. Interfacial bubbles introduce strain in the top hBN layer, which hinders this growth. Consequently, the organic nanostructure masks, and therefore the final graphene nanoribbon network, preferentially form over pristine unstrained regions, avoiding the bubbles. Hence, the Gr NR network is formed away from these potential charge traps.

Changes made: This issue is now clearly discussed in the main text, explaining how our method allows minimizing the influence from interfacial air bubbles. The SI has been updated with a new Figure S3, providing direct evidence by AFM topography imaging of the nanostructure's assembly, avoiding the bubbles.

Figure S3. ... (d-g) AFM topography images of the nanoribbon networks before the RIE step (scale bar 500 nm, z scale 30 nm). Observed bubbles are characteristic interfacial traps of water-air mixture within the layers of the van der Waals hetero-stack (hBN/graphene/hBN). Bi-axial strain of the top layer induced by the bubbles hinders the self-assembly of the organic nanostructures. Consequently, the organic nanostructures tend to form between the bubbles or tend to terminate on the bubbles.

R2Q4: The authors could consider conducting experiments on wider graphene devices encapsulated by hBN. Could they comment on how the width of the graphene affects the hysteresis characteristics?

Authors' response: We thank the reviewer for bringing up this excellent point. In flakes (width commonly above 2 μm), this effect is not observed, unless the current is passed along the flake's edge [10.1021/acs.nanolett.8b00797]. Otherwise, an alternative approach to investigate the impact of the ribbon width on the V_H would be to use EVU lithography or self-assembly of block copolymers. Although such a systematic experimental investigation is beyond the scope of this work, we performed molecular dynamics simulations to elucidate how NR's width affects hysteresis.

Changes in nanoribbon width should affect the magnitude of the hysteresis window, but neither the nature of the observed effect nor the conclusions regarding the layer number and temperature dependence.

Reviewer 1 raised similar concerns in their questions 1 and 2. Therefore, the following part of the answer is repeated from Reviewer 1 Question 2:

We have estimated the width limit and discussed this issue in more detail in the revised manuscript. From MD simulations of polarized dipoles, both for 1L and 2L ribbons, we have extracted the dipole-induced fields perpendicular to the basal plane of graphene and their dependence along the x direction of the ribbon width. The fields induced by only one polarized edge are illustrated to show that, from the MD model, we obtain the theoretically expected $1/x^2$ dependence (added in Figure S19).

There are two scenarios to consider:

- As the ribbon width reduces, at some point, the field induced by one row of the polarized edge dipoles would start to affect the dipoles at the other edge. The dipoles from the two edges of a ribbon would effectively try to flip each other, as the edge dipole-induced field has the opposite sign of the external field by the gate that introduces their polarization. From the MD simulations, we estimate that below a 10 nm width, the field induced by one dipole line (edge) starts to affect the polarization of the opposing edge. Below 5 nm, the dipole field would be sufficient to overcome the external field and would likely result in an antiparallel state. Consequently, we expect **a reduction in the hysteresis effect for widths below 10 nm**.
- Looking into the other extreme, as the ribbon width increases, more "bulk" properties of graphene would be observed in the electrical response, and the hysteresis would reduce. Again, from the MD simulations, we estimate that **the widths above 100 nm GrNRs would give rather a negligible contribution from the edges** to the electrical response. In general, the wider the ribbons, the lower the impact of the edge-induced fields.

Judging from the dipole-induced field distribution from our MD simulations, we expect that the nanoribbon width in the range of **10 nm is optimal for maximizing the hysteresis window** with respect to the total range of the field that can be applied by the gate. Additionally, optimizations with high-k dielectrics, reduced gate thickness, and the use of dual-gate geometries would enable further enhancement of the water switch's effectiveness.

Changes made: The impact of the width of the ribbons is discussed in the main text, and is referenced to the added section in the SI Figure S7 and Figure S19 where, respectively, the statistics of V_H for all the devices (see also the reply to reviewer 1 question 6) and additional MD results on the field decay along the ribbon width direction are provided. Estimates of the expected optimal ribbon width, along with other potential improvements of the device architecture, are also discussed in the main text.

Figure S19: MD simulations of the field induced in y -direction (perpendicular to the basal plane) by the polarized dipoles ($E_{\text{ext}} = 2$ V/nm in the positive y -direction, 300 K) considering only one ribbon edge, in the case of 1L ribbon (a-c) and 2L ribbon (d-f). (a,d) Contour maps presenting the surface charge density. Electric field streamlines are superimposed to indicate the direction of electrostatic influence. The mirror-image placement of charges ensures lateral symmetry, and the color-coded regions emphasize strong confinement effects in the near-surface zone. (b,e) corresponding E_y field strength map (semi-log scale). (c,f) Intensity of the dipole induced fields (solid blue lines) along the x -direction at 0.2 nm above the graphene plane, in log-log scale. A reference x^{-2} trend line (dashed orange) illustrates the asymptotic decay expected from a 1D line of dipoles, confirming consistency with analytical predictions.

Reviewer #3 (Remarks to the Author):

This manuscript reports on the role of water molecule dynamics in the ferroelectric response of graphene nanoribbon (Gr NR) devices. Through an analysis of the dependence on the number of graphene layers and temperature, it reveals the stabilizing effect of the collective behavior of water molecules on the ferroelectric effect. To enhance the quality of the manuscript, minor revisions should be considered. All recommendations are listed below.

Authors' response: We thank the reviewer for the positive evaluation and constructive recommendations. We have addressed all the reviewers' comments as described below.

R3Q1: The manuscript involves the preparation of multilayer graphene nanoribbons, water molecule adsorption, and temperature-dependent measurements. However, it does not clarify how experimental consistency was ensured, particularly regarding the reproducibility of experiments under varying temperature and electric field conditions. The authors should demonstrate the consistency of experimental results across different devices.

Authors' response: We thank the referee for bringing up this important point. Following the reviewer's advice, we took care of this question, addressing it in two ways:

- 1) We conducted and included a comprehensive statistical analysis of all 22 Gr NR devices used in this study, comprising graphene thicknesses from 1L to 5L. These results are included in the revised SI, new Figures S5 and S7. Although there is some expected scatter in the absolute mobility and hysteresis values due to the random nature of the nanostructure self-assembly process, the statistical results show that the trends and performance reported are consistent across all devices.
- 2) To clarify how we maintained experimental consistency, we expanded the SI Methods section to include all fabrication conditions. These conditions adhered to the standards commonly found in nanofabrication labs, with each device's fabrication parameters closely monitored. All electrical measurements were conducted using systematically defined measurement routines.

Changes made: The SI Methods now state additional information on the experimental procedures, especially focusing on ensuring comparable experimental conditions between the different devices. These include:

- Holding times in the HWE setup before the growth to ensure stable growth temperatures (45 min).
- Variations of the temperatures for the HWE experiments are also stated.
- Holding times in the electrical setup to ensure stable temperature during variable temperature electrical measurements; 10 min for the flow LN2 cryostat and the 77 K to 425 K range, and 60 min for the closed-cycle LHe cryostat and 4.2 K measurements.
- Temperature variations are specified for the electrical setups.
- Vacuum variations are specified for the electrical setups.
- Device annealing time and temperature are stated (see also reviewer 1 question 5)
- Relative humidity exposure after RIE and after vacuum annealing is now clearly defined (humidity values and exposure time) - (see also response to reviewer 1 questions 3 and 5)

SI Figures S5 and S7 are added to provide an overview of the peak apparent electron mobility and a summary of the hysteresis window (V_H). The addition is also briefly discussed in the main text (see also reviewer 2 question 1, and reviewer 1 question 6).

Figure S5: Device statistics for the linear apparent electron mobility as a function of the number of graphene layers $\mu_{lin}(L)$. The values are extracted as peak mobilities from the electrical transfer curves recorded at 300 K ($2 \cdot 10^{-2}$ mbar). The total of 22 devices that were used in the study is presented. Hexagons correspond to the flake-FETs prior to the RIE step (step 5 in the fabrication scheme – Figure S1), diamonds represent Gr NR-FETs etched using SF_6 RIE, and circles using O_2 RIE. Mobilities are extracted in all cases using a linear regime approximation and a parallel capacitance model. Fringing capacitance model for the NR-FETs results in about one order of magnitude lower values [S2]. Length and width of each device were determined via a combination of the optical micrographs and AFM topography images. The larger uncertainty for the Gr NR-FETs originates from the uncertainty of the NR widths measured by the AFM.

Figure S7. Statistics of the hysteresis window (V_H) considering the entire set of devices used in the study (16 O_2 RIE and 6 SF_6 RIE FETs). Red circles present O_2 etched devices and purple diamonds present SF_6 etched ones. Shaded areas serve as a guide to the eye to indicate the trends and the device-to-device scattering of the parameters. The data is extracted from the electrical transfer curves measured at 300 K and under $2 \cdot 10^{-2}$ mbar. Each data point with its error bar is extracted as a mean and standard deviation from five transfer curves. (a,b) V_H is scaled by the total V_{SG} sweeping range (V_{SG-tot}), as a function of the number of graphene layers (a) and as a function of V_{SG-tot} (b). (c) presents $V_H(V_{SG-tot})$ without scaling. Likely, the main source for the device-to-device scatter is the variation in the width of the nanoribbons in the self-assembled networks.

R3Q2: The experimental results in the manuscript show good consistency with the molecular dynamics simulation results for multilayer graphene nanoribbons (Gr NRs), but there are certain discrepancies in the case of monolayer graphene nanoribbons. For example, the opening of the hysteresis window shown in Fig. 3 (a) is much less clear as compared to the experimental results (Figure S4). The authors explained this by introducing the possible more complex interfaces due to hBN encapsulation, please explain in more detail.

Authors' response: We thank the reviewer for this question, and agree that this issue was not clearly explained in the original manuscript. After the reviewer's comment, we have now provided more details about the discrepancy between mono-layers and multi-layers in the revised manuscript, clarifying that the collective motion of water and cluster formation is not excluded for the mono-layers (please see also our response to reviewer 1's question 7). We attribute these differences to the stabilizing effect of the hBN encapsulation, which was not included in the MD simulations due to its computational complexity. In short, the hBN encapsulation with the polar B-N bonds provides an additional electrostatic potential near the mono-layer's edge that favors the water molecules' collective behavior. Another consequence of the hBN encapsulation is the spatial confinement it creates, hindering the motion of water molecules. Since these effects were not captured in the MD simulations, there is a discrepancy with the actual experimental results.

Changes made: The expected influence of the hBN caps on the V_H difference between the experiment and the MD simulation is now discussed in more detail.

R3Q3: The time scale of the molecular dynamics simulations (4 ns) in the manuscript significantly differs from the experimental time scale (100 ms). In what way could this substantial discrepancy in time scales affect the accuracy of the simulation results?

Authors: We thank the referee for pointing out this issue that was insufficiently addressed in the original manuscript. The different time scales are a known challenge when comparing MD simulations with actual experiments. In our case, the system has more time to relax, meaning that upon the application of the bias, stronger polarization of the dipoles would be expected in our experiments. To compensate for this in the simulations, we applied higher fields than achievable by the experiments to speed up the system dynamics and probe the stability of the water clusters and their collective response. From this perspective, we had to limit the lower temperature for the model to 250 K, as at lower T , a much longer time would be needed for the relaxation. Therefore, and considering that the MD model is idealized and a much smaller system than the one probed in the experiment, we relied on these simulations to reveal the physical mechanism, but not for a direct quantitative match with the experimental data. The MD simulations proved to be a valuable tool for identifying trends that support our main experimental findings, particularly the temperature dependence in monolayers versus the independence in bilayers, thus confirming the mechanism we proposed.

Changes made: The discussion on the impact of the differing time scales between the experiments and the MD simulations is now expanded. We clearly state that, due to these discrepancies, the absolute values for the remnant fields and the hysteresis opening cannot be directly compared to the experimental results in a quantitative manner. Instead, the temperature trends are effectively captured and used for the comparison with the experiment, validating the physical mechanism we proposed.

R3Q4: The ferroelectric effect in multilayer graphene nanoribbons (2-5 layers) remains nearly unaffected at temperatures up to 450 K, whereas the ferroelectric effect in monolayer graphene nanoribbons exhibits significant temperature dependence. While the authors attribute this phenomenon to the absence of collective water molecule behavior in monolayer graphene nanoribbons, they do not explore the mechanistic differences in temperature stability between graphene nanoribbons of varying layer numbers.

Authors' response: Following the reviewer's advice, we have added a new Figure S13 to the revised SI to better illustrate the difference in temperature stability, showing the self-stabilizing collective state in multi-layers. This is a critical mechanistic difference that is based on the formation of water bridges in multi-layers ($L > 2$), which is physically impossible in mono-layers. These bridges act as anchoring sites for water molecules, enabling the formation of larger and more robust water clusters compared to the case of mono-layers. Consequently, the strong intermolecular interactions mediated by these water bridges give rise to a self-stabilizing state depicted in Figure S13, which can be seen as a deepening of the potential energy well of the polarized state. This provides an additional energy barrier for depolarization in multi-layers, making the system resistant to thermal energy and thereby providing temperature-independent behavior to the hysteresis, as observed in the $V_H(T)$ trend for all multi-layered devices we investigated. In contrast, in mono-layers that cannot support the formation of these water bridges, the stabilizing anchoring effect is missing, making the water clusters at the edges less stable and their polarization more easily affected by thermal energy, which explains the observed temperature dependence.

Regarding the mechanistic differences for a varying number of layers, in Figure 1f of the revised manuscript, we now specify the number of layers for each curve. The critical observation is that these plots vs. temperature are flat and overlapping, with no discernible trend, in contrast to the results for mono-layers in Figure 1e, which show temperature-dependent behavior. This is why the mechanistic difference we focus on is between the mono-layer case and the multi-layer $L > 2$ case.

Changes made: We now discuss in more detail these mechanistic differences in temperature stability. A conceptualization of the bi-stable system and the differences between such a system with and without the effects of the inter-particle interactions and self-stabilization are mentioned in the main text and are discussed in more detail in the SI Figure S13.

R3Q5: It would be valuable if the authors could further clarify the potential benefits of designing water-switch-based devices for in-memory computing and synaptic circuit applications.

Authors' response: We thank the reviewer for raising this valuable point, which is actually one of the directions that our team is currently working on. We envision the non-volatile field-programmable nature of the edge adsorbed water switch as promising for computing applications in two configurations:

First, as robust memristors with non-volatile synaptic weights. This could be achieved by maximizing the hysteresis effect in a dual gate (top and bottom) geometry architecture with high- k gate dielectrics to enhance the V_H range while reducing operating voltages V_{SG} . Further engineering of narrower and pitch-controlled ribbons would enhance the on/off ratio, allowing memristive switching as highly tunable synaptic elements controlled by the edge water dipoles.

Example structure of a dual-gate Gr-NR FET.

Second, one could exploit the nanoribbon network as a programmable ferroelectric electrode in a vertical field effect transistor (V-FET) configuration with an ambipolar 2D semiconductor as a channel. In this case, the dipolar field of the water molecules would non-volatily dope the 2D semiconductor channel, allowing **switching between a p -type, n -type, and the off state**, making possible a device that acts as a multi-state memory cell and a logic gate. A network of these could make a truly reconfigurable circuit, which is a key goal for in-memory computing.

Example structure of a V-FET with Gr-NR electrode

Changes made: The possible geometries for future integration of the water switch into computing-in-memory elements are briefly addressed in the Discussion section of the revised manuscript.

R3Q6: There are minor typographical errors in the manuscript, such as "VH" on line 87 not being italicized. Please ensure that all content adheres to standard formatting conventions.

Authors' response: We thank the reviewer for the careful reading of our work and for noticing this omission. We have performed a thorough review of the entire manuscript and SI to correct all typos and ensure consistent formatting of all variables, constants, and abbreviation definitions.

Changes made: Corrected typographical errors are highlighted in the marked-up versions of the revised manuscript and SI by red font color.

REVIEWERS' COMMENTS NCOMMS-23-18909B

Reviewer #1 (Remarks to the Author):

The Authors exhaustively replied to all raised concerns and added significant new data and extra discussion in the revised version of the manuscript. I therefore recommend the publication of this manuscript as is.

Authors' response: We're thankful to Reviewer #1 for their time and effort in evaluating this work and for the positive feedback following the revisions.

Reviewer #2 (Remarks to the Author):

Because transport experiments probe the overall average electrical properties of a sample, it is inherently challenging to disentangle contributions that arise solely from the edge. As a result, arguments by authors cannot yet be regarded as completely conclusive. However, it is also true that the authors provided reasonable answers and persuasive arguments to the various concerns I raised.

In my view, the manuscript is suitable for publication at this stage, allowing the broader readership to assess its merits.

Authors' response: We're thankful to Reviewer #2 for their critical feedback and evaluation of our work, which helped make it significantly better.

Reviewer #3 (Remarks to the Author):

The authors have addressed my comments clearly, I have no more questions.

Authors' response: We're thankful to Reviewer #3 for their time and effort in evaluating this work and contributing to its increased quality and clarity in the revised version.